# CORRELATED PROXIES: A NEW DEFINITION AND IMPROVED MITIGATION FOR REWARD HACKING

**Cassidy Laidlaw**[*]    **Shivam Singhal**[*]    **Anca Dragan**
Department of Electrical Engineering and Computer Science
University of California, Berkeley
Berkeley, CA 94709, USA
{cassidy_laidlaw,shivamsinghal,anca}@berkeley.edu

## ABSTRACT

Because it is difficult to precisely specify complex objectives, reinforcement learning policies are often optimized using proxy reward functions that only approximate the true goal. However, optimizing proxy rewards frequently leads to reward hacking: the optimized reward function ceases to be a good proxy and the resulting policy performs poorly with respect to the unspecified true reward. Principled solutions to reward hacking have been impeded by the lack of a good definition for the problem. To address this gap, we introduce a definition of reward hacking based on the correlation between proxy and true rewards for states and actions seen by a "reference policy" that breaks down under optimization. We show that this definition captures reward hacking behavior across several realistic settings, including in reinforcement learning from human feedback (RLHF). Using our formulation, we show theoretically that regularization to the reference policy can effectively prevent reward hacking. While the current practice in RLHF applies a KL penalty between action distributions for this purpose, our theory suggests regularizing the $\chi^2$ divergence between the policies' occupancy measures can be more effective. We intuitively show the benefits of this type of regularization and demonstrate that it better mitigates reward hacking in practice across four realistic settings, including RLHF. Our code is available at https://github.com/cassidylaidlaw/orpo.

## 1    INTRODUCTION

A major challenge for the designers of goal-oriented AI systems is specifying a reward function that robustly captures their goals and values. Manually designing reward functions is difficult due to the ambiguities and complexity underlying real-world scenarios (Ibarz et al., 2018). An alternative is to learn reward functions from human data (Sadigh et al., 2017; Jeon et al., 2020), but these often fail to generalize outside the distribution of behavior seen during training (McKinney et al., 2023; Tien et al., 2023). Thus, a learned or hand-specified reward function is often just a *proxy* for the true reward underlying the system designer's intent. Misalignment between the two objectives can lead to *reward hacking*: a learned policy performs well according to the proxy reward function but not according to the true reward function (Russell et al., 2010; Amodei et al., 2016; Pan et al., 2022; Skalse et al., 2022). A reward hacking policy's behavior is often undesirable and can be especially catastrophic when deployed in safety-critical scenarios, such as autonomous driving (Krakovna et al., 2019; Turner et al., 2019; Knox et al., 2022). Unfortunately, reward hacking is a common phenomenon (Krakovna, 2018) and has harmful effects in the real world (Lum & Isaac, 2016; Corbett-Davies et al., 2017; Obermeyer et al., 2019; Milli et al., 2021; Franchi et al., 2023; Kleinberg et al., 2023).

The ideal solution to prevent reward hacking would be to perfectly align the specified proxy and unknown true reward; however, in many domains, this is impossible to achieve. For example, imagine trying to design a reward function for a self-driving car. It would need to capture multiple factors: the arrival time, passenger comfort, compliance with traffic laws, and various other considerations—many of which are difficult to robustly measure and would need to be carefully weighted against each other (Knox et al., 2022). In practice, reward hacking can occur even with significant reward engineering

---

[*]Equal contribution.

Figure 1: We present a new characterization of reward hacking and a method for preventing it. We define a proxy reward function as one that correlates with an unknown true reward function for state-action pairs sampled from some reference policy. However, optimizing the proxy alone can lead to a breakdown in the correlation and worse true reward than the reference policy. We show theoretically and empirically that optimizing the proxy with $\chi^2$ occupancy measure regularization to the reference policy can allow outperforming the reference policy under the unknown true reward.

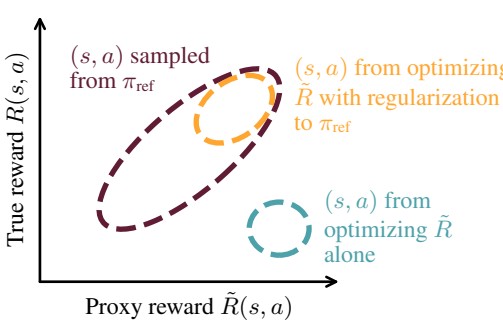

efforts. This is evident in the case of recommender systems, where hand-designed reward functions have led to adverse outcomes in terms of user experience (Stray et al., 2022).

Since proxy reward functions for complex tasks are nearly always misspecified in practice, what can be done to avoid reward hacking? There is a lack of principled solutions for preventing reward hacking, stemming from a more fundamental problem: *defining* reward hacking in a formal sense that captures realistic cases. In particular, it is difficult to characterize what makes a proxy reward function "good". We argue that proxies are chosen because they seem to capture the true objective under some reference distribution of behavior. To formalize this, we define a proxy reward as one that *correlates* with the true reward function under the distribution of states and actions visited by a *reference policy*. Then, we define reward hacking in this setting as when optimizing a correlated proxy reward leads to a policy obtaining lower true reward than the reference policy (Figure 1). This captures several intuitive cases of reward hacking across realistic environments: traffic control, pandemic mitigation, blood glucose regulation, and reinforcement learning from human feedback (RLHF) (Figure 2).

Furthermore, our definition motivates a method for avoiding reward hacking during policy optimization by regularizing the optimized policy to be similar to the reference policy. Specifically, we find that optimizing the proxy reward minus a regularization term provides a *provable* lower bound on improvement in true reward. The amount of regularization needed increases as the correlation between the proxy and true rewards becomes weaker. In practice, RLHF already uses this form of regularization through a KL divergence penalty to the reference policy (Stiennon et al., 2020; Bai et al., 2022), and Theorem 5.1 provides theoretical justification for why this works. However, our results suggest an important modification: rather than penalizing the KL divergence between the policies' *action distributions* as currently done in RLHF, we should instead penalize the chi-squared ($\chi^2$) divergence between the policies' *occupancy measures*. We compare both OM to AD regularization and $\chi^2$ to KL divergence, presenting intuitive reasons why $\chi^2$ OM divergence may be a better regularization target in Figures 3 and 4.

Based on our theoretical results, we empirically investigate the benefits of using $\chi^2$ occupancy measure regularization to prevent reward hacking. Following previous work (Ho & Ermon, 2016), we implement OM regularization using a discriminator network that approximates the OM divergence between policies. We evaluate our approach in multiple reward hacking benchmark environments (Pan et al., 2022). Our results demonstrate that training with occupancy measure regularization leads to better performance under the unseen true reward function in all of the environments, validating our theoretical results. In contrast, we find that it is difficult to tune AD regularization in some environments to both prevent reward hacking and allow meaningful improvement over the reference policy. Furthermore, regularization with $\chi^2$ divergence tends to be more robust to the choice of regularization coefficient and often achieves higher true reward than regularizing with KL divergence.

Our main contributions can be summarized as follows:

1. We provide a new formal definition of reward hacking that captures realistic case studies.
2. Using our definition, we establish that optimizing a correlated proxy reward with $\chi^2$ occupancy measure (OM) regularization leads to a provable improvement in the unknown true reward function.
3. We show how to practically implement $\chi^2$ OM regularization and demonstrate that it outperforms the current standard for preventing reward hacking via action distribution (AD) regularization.

## 2 RELATED WORK

**Reward hacking.** Some prior works establish theoretical models of reward hacking as a special case of Goodhart's Law (Goodhart, 1984; Leike et al., 2018; Manheim & Garrabrant, 2019; Krakovna, 2019; Skalse et al., 2022; Ngo et al., 2023; Fluri et al., 2024; Kwa et al., 2024). Krakovna (2018) catalog many examples of reward hacking. Pan et al. (2022) categorize different types of reward misspecification and relate optimization power to reward hacking. See the end of Section 4 for a comparison of our definition of reward hacking with previous ones.

**Safe reinforcement learning.** In the context of reward hacking, regularizing policies to be similar to an offline policy based on their action distribution KL divergence was proposed by Stiennon et al. (2020) and has since been widely employed in the context of optimizing LLMs using RLHF (Ouyang et al., 2022; Bai et al., 2022; Glaese et al., 2022). KL regularization for RLHF has been further studied by Vieillard et al. (2021), Gao et al. (2022), and Korbak et al. (2022). Nika et al. (2024) propose a type of occupancy measure regularization in RLHF but it is limited to deterministic MDPs, while we study general stochastic MDPs. Some alternative approaches to avoid reward hacking include quantilizers (Taylor, 2016), "mild" optimization (Taylor et al., 2020), and impact regularization (Turner et al., 2020). While constrained RL can prevent the misbehavior of agents that optimize flawed reward functions (Dalal et al., 2018; Chow et al., 2019; Zhang et al., 2020; Roy et al., 2022), it simply shifts the difficulty of designing a reward function to specifying a set of constraints and weights. Robust RL usually considers a misspecified transition model, but some work has explored misspecified reward functions (Derman et al., 2021; Gadot et al., 2024). Other proposals to address the reward specification problem attempt to infer the true reward function based on the given proxy reward function, environment context, and/or feedback from humans (Hadfield-Menell et al., 2017; Reddy et al., 2020; Lee et al., 2021). Gleave et al. (2021) and Skalse et al. (2024) have previously studied quantifying the similarity of reward functions.

**Other divergences for regularization in RLHF.** Some work has explored using divergences other than KL divergence for regularization in RLHF. Go et al. (2023) investigate using other $f$-divergences for aligning language models. Wang et al. (2023a) explore generalizing direct preference optimization (DPO) (Rafailov et al., 2024), a type of RLHF that leverages offline RL, to other $f$-divergences. The most relevant work to ours in this area is Huang et al. (2024), which also proposes using $\chi^2$ divergence as a more theoretically grounded alternative to KL divergence for regularization in RLHF. Our work differs from theirs in several key aspects: we address reward hacking in general rather than focussing specifically on learning from preference data; we use online RL instead of their DPO-like algorithm; and, we validate our approach through experiments in realistic environments while their work is primarily theoretical.

**Applications of occupancy measure regularization.** Occupancy measure regularization and optimization have been used for a variety of purposes, including imitation learning (Ho & Ermon, 2016) and efficient exploration Kang et al. (2018); Hazan et al. (2019); Lee et al. (2020); Nedergaard & Cook (2023). Various types of distributional regularization are used in model-based RL since learned models may not generalize out-of-distribution (Yang et al., 2022).

**Offline reinforcement learning.** Our work bears a superficial resemblance to prior work in offline RL, where algorithms commonly use occupancy measure or action distribution-based regularization to constrain the learned policy within the training data distribution (Fujimoto et al., 2019; Lee et al., 2022; Mandal et al., 2023; He, 2023; Cheng et al., 2022; Rashidinejad et al., 2023; Xie et al., 2023). However, the settings are fundamentally different: while offline RL is limited by a lack of coverage in the training data, the difficulty in our setting is that the reward function is misspecified. While it might be possible to avoid reward hacking by using offline RL algorithms, we leave this to future work and focus on regularization-based approaches in the online RL setting.

## 3 PRELIMINARIES

To study reward hacking, we consider the setting of an infinite-horizon Markov decision process (MDP). An agent takes actions $a \in \mathcal{A}$ to transition between states $s \in \mathcal{S}$ over a series of timesteps $t = 0, 1, 2, \ldots$. We assume that $\mathcal{S}$ and $\mathcal{A}$ are finite for simplicity, but our results can easily generalize to infinite state or action spaces. The first state $s_0$ is sampled from an initial distribution $\mu_0(s)$, and when an agent takes action $a_t$ in $s_t$ at time $t$, the next state $s_{t+1}$ is reached at timestep $t + 1$ with

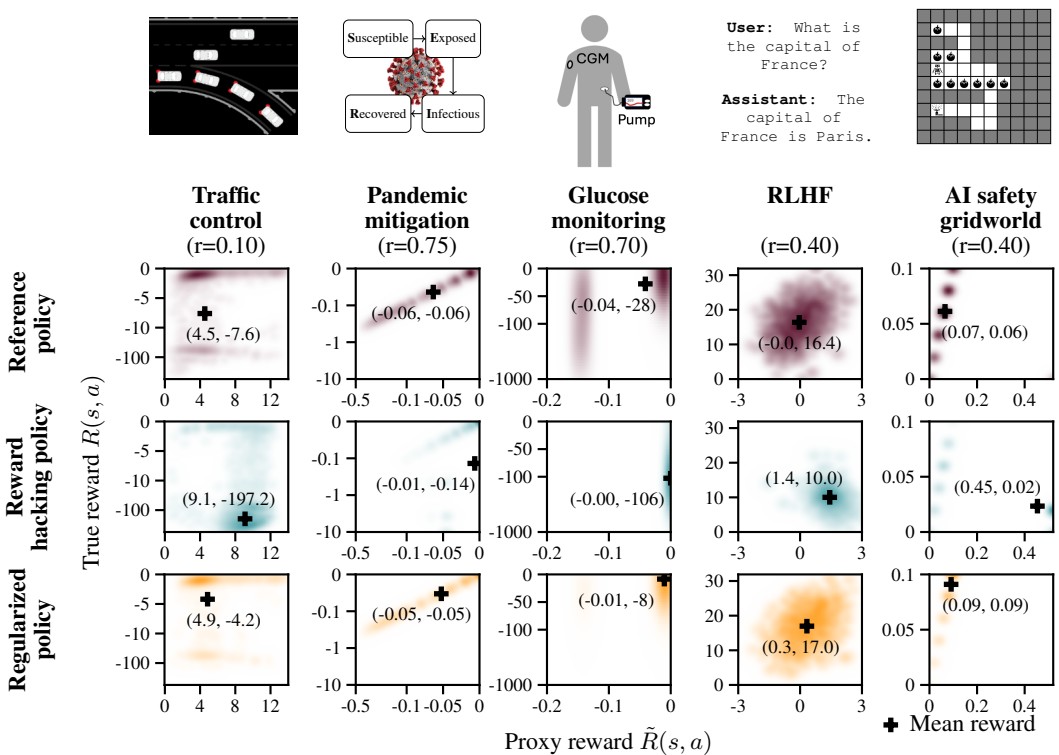

Figure 2: Our definition of reward hacking successfully describes reward hacking behavior in four realistic environments and an illustrative gridworld. The top row shows the distribution of proxy and true reward values for state-action pairs sampled from a domain-appropriate reference policy for each environment; in all environments, the proxy and true rewards are correlated. However, as shown in the middle row, this correlation breaks down if the proxy is optimized via RL, and the true reward achieved is lower than that of the reference policy, which we characterize as reward hacking. In line with our theoretical results, the bottom row shows that RL with occupancy measure regularization to the reference policy can prevent reward hacking while enabling an increase in true reward.

transition probability $p(s_{t+1} \mid s_t, a_t)$. The agent aims to optimize a reward function $R : \mathcal{S} \times \mathcal{A} \to \mathbb{R}$, and rewards are accumulated over time with discount factor $\gamma \in [0, 1)$. A policy $\pi$ maps each state $s$ to a distribution over actions to take at that state $\pi(a \mid s)$. We define the (normalized) *return* of a policy $\pi$ under a reward function $R$ as

$$J(\pi, R) = (1 - \gamma) \mathbb{E}_\pi \left[ \sum_{t=0}^{\infty} \gamma^t R(s_t, a_t) \right] \tag{1}$$

where $\mathbb{E}_\pi$ refers to the expectation under the distribution of states and actions induced by running the policy $\pi$ in the environment.

We define the *state-action occupancy measure* $\mu_\pi$ of a policy $\pi$ as the expected discounted number of times the agent will be in a particular state and take a specific action:

$$\mu_\pi(s, a) = (1 - \gamma) \mathbb{E}_\pi \left[ \sum_{t=0}^{\infty} \gamma^t \mathbb{1}\{s_t = s \wedge a_t = a\} \right]. \tag{2}$$

The standard approach to solving an MDP is to find a policy $\pi$ that maximizes its return $J(\pi, R)$. However, as we discussed in the introduction, an AI system designer might optimize $\pi$ using a learned or hand-specified *proxy* reward function $\tilde{R}$ that imperfectly captures the *true* reward function $R$. A precise definition of reward hacking would help us to better understand and mitigate the problem. Intuitively, reward hacking occurs when when optimizing the proxy reward $J(\pi, \tilde{R})$ of a policy $\pi$ ultimately leads to low true reward $J(\pi, R)$. However, formalizing this intuition is surprisingly challenging.

**Desiderata for a definition of reward hacking.** To understand why defining reward hacking is difficult, consider the case study of RLHF. In RLHF, optimizing a learned reward function over LLM outputs eventually leads to the LLM producing nonsensical responses Stiennon et al. (2020). This clearly satisfies our informal definition of "optimizing the proxy makes the true reward decrease."

However, what if we were to optimize $\tilde{R}(s,a) = -R(s,a)$? In this case, optimizing $\tilde{R}$ also makes the true reward go down, but arguably this is not reward hacking because $\tilde{R}$ was not a good "proxy" in the first place. Thus, a good definition of reward hacking should distinguish between "reasonable" proxies and reward functions that are clearly unrelated to the true reward function.

In the example of RLHF, when optimization leads to nonsensical outputs, we can clearly identify that the proxy has been "hacked." However, what if optimizing the proxy produced mediocre but not obviously bad outputs? Even if optimizing the proxy does not lead to near-optimal true reward, we wouldn't necessarily classify this behavior as reward hacking. Thus, a good definition of reward hacking should specify a threshold for the true reward: if the true reward falls below the threshold, reward hacking has occurred, and above it, it has not.

To summarize, we argue that a good definition of reward hacking should satisfy two desiderata:

1. Characterizing what makes a proxy reward "reasonable" to optimize in the first place.
2. Defining a threshold for true reward below which a policy is reward hacking.

After presenting our definition in the next section, we compare to other definitions using these criteria.

## 4 DEFINING REWARD HACKING

Despite the difficulties in formalizing reward hacking, we argue that both of our desiderata for a definition can be met by defining reward hacking with respect to a *reference policy*. This approach enables both a precise definition of proxy rewards and a natural true reward threshold for determining when reward hacking is occurring.

**Characterizing proxy rewards.** To find a realistic definition of a good proxy reward, consider the process by which system designers create proxies. If they are hand-specified, we argue that designers usually imagine a distribution of behavior and design a reward that captures the objective under that distribution. For example, we study a traffic control simulator (Lopez et al., 2018; Vinitsky et al., 2018; Wu et al., 2022; Pan et al., 2022) where a small number of autonomous cars help to regulate traffic among a larger population of human drivers. In this case, a designer might choose average vehicle speed as a proxy for traffic flow; this is a reasonable proxy because under the distribution of typical human driving behavior, higher speeds are associated with better traffic conditions. We can formalize this intuition by requiring that the proxy and true rewards be *correlated* over states and actions sampled from a *reference policy*:

**Definition 4.1** (Correlated proxy reward). An $r$-correlated proxy reward $\tilde{R}$ with respect to a reference policy $\pi_{\text{ref}}$ is one that has a correlation of $r > 0$ with the true reward for state-action pairs sampled from the reference policy:

$$\mathbb{E}_{\mu_{\pi_{\text{ref}}}} \left[ \left( \frac{\tilde{R}(s,a) - J(\pi_{\text{ref}}, \tilde{R})}{\sigma_{\tilde{R}}} \right) \left( \frac{R(s,a) - J(\pi_{\text{ref}}, R)}{\sigma_R} \right) \right] = r,$$

where $\quad \sigma_{\tilde{R}}^2 = \mathbb{E}_{\mu_{\pi_{\text{ref}}}} \left[ \left( \tilde{R}(s,a) - J(\pi_{\text{ref}}, \tilde{R}) \right)^2 \right] \quad$ and $\quad \sigma_R^2 = \mathbb{E}_{\mu_{\pi_{\text{ref}}}} \left[ \left( R(s,a) - J(\pi_{\text{ref}}, R) \right)^2 \right]$

are the variances of proxy and true rewards under the reference policy.

To consider an application of our definition, let's return to the example of our traffic environment, where we define the true reward function as the negative total commute time for all vehicles. Using an autonomous vehicle policy based on a common model of human driving behavior as $\pi_{\text{ref}}$, we plot the distribution of true and proxy rewards in the top-left corner of Figure 2. The clear correlation that we observe between the two reward functions—higher average speed corresponding to lower commute times and vice versa—confirms that average speed is a correlated proxy reward according to Definition 4.1.

Definition 4.1 offers key advantages over past formalisms. First, it avoids too strongly constraining proxy rewards by allowing the proxy and true reward functions to diverge arbitrarily at some state-action pairs when those pairs have low or zero occupancy measure under the reference policy. Second, it encompasses both *learned* proxy rewards and hand-specified ones; see Lemma A.8 for a discussion of how learned rewards are generally correlated proxies.

**Choosing a threshold for reward hacking.** As discussed in Section 3, defining reward hacking requires specifying a threshold of true reward below which policy behavior is considered poor enough

to be classified as reward hacking. Given that we already characterize proxy rewards with respect to a reference policy, this same reference policy provides a natural baseline for evaluating a policy that optimizes the proxy: if optimizing the proxy leads to worse true reward than the reference policy achieves, then the system designer may be better off simply using the reference policy.

**Definition 4.2** (Reward hacking). Suppose $\tilde{R}$ is an $r$-correlated proxy with respect to $\pi_{\text{ref}}$ (Definition 4.1). Then we say *reward hacking* occurs when a policy $\pi$ optimized for $\tilde{R}$ has lower true reward than the reference policy $\pi_{\text{ref}}$, i.e., when $J(\pi, R) < J(\pi_{\text{ref}}, R)$.

If an optimal policy for $\tilde{R}$ exhibits reward hacking, then we say that $\tilde{R}$ is a *hackable correlated proxy*:

$$J(\pi, R) < J(\pi_{\text{ref}}, R) \qquad \text{for some} \qquad \pi \in \arg\max_{\pi} J(\pi, \tilde{R}).$$

For example, in the traffic control environment, optimizing average speed as a proxy leads to autonomous vehicles blocking highway on-ramps, a strategy that increases overall average speed by allowing highway traffic to move freely. However, the true reward becomes extremely low because commute times for the cars on the on-ramp are arbitrarily long. This policy is thus reward hacking under Definition 4.2 since the true reward is lower than that of typical human driving.

**Verifying our definition experimentally.** To test whether Definition 4.2 accurately captures intuitive cases of reward hacking in practice, we consider four realistic environments and an illustrative gridworld. In addition to the aforementioned traffic simulator, we consider pandemic mitigation and glucose monitoring environments, all of which were originally studied by Pan et al. (2022) as examples of reward hacking. PandemicSimulator (Kompella et al., 2020) uses a specialized SEIR (susceptible, exposed, infected, recovered) infection model to simulate the COVID-19 pandemic among a population; the policy controls the level of lockdown restrictions placed on the population by observing the results of testing. The proxy reward omits the political cost of certain decisions. SimGlucose (Man et al., 2014) is based on an FDA-approved simulator of Type 1 Diabetes patients in which a policy monitors glucose levels and administers insulin. The true reward captures patient health while the proxy reward prioritizes reducing the monetary cost of insulin.

We also study RLHF, in which LLMs are optimized based on a reward function learned from human preferences. Following Gao et al. (2022) and Coste et al. (2024), we use a large (more robust) reward model as the true reward function and a smaller (less robust) one as the proxy. Finally, as an interpretable and illustrative example, we include the tomato-watering AI safety gridworld from Leike et al. (2017). Besides the gridworld, all of our environments reflect complex, realistic tasks with large or infinite state state spaces. We construct a natural reference policy for each of the five environments; see Appendix C for more details about the environments and reference policies.

The top row of Figure 2 shows the distribution of true and proxy reward values for state-action pairs sampled from these reference policies in each environment. We find that in all cases, the proxy rewards correlate with the true reward, satisfying our definition of a correlated proxy. Furthermore, when optimizing the proxies, we see that the true reward drops significantly compared to the reference policy. Thus, these intuitive cases of reward hacking are captured by Definition 4.2.

**Comparison to other definitions of reward hacking.** Skalse et al. (2022) define reward hacking as an increase in proxy reward accompanied by a drop in true reward. This definition does not satisfy our first desideratum of requiring the proxy to be a reasonable optimization target. Also, the threshold of *any* decrease in the true reward being considered reward hacking seems overly restrictive; in many cases, optimizing a proxy reward will not lead to a perfect optimization of the true reward. Kwa et al. (2024) define the phenomenon of "catastrophic Goodhart" as when optimizing a proxy with low expected error under some base policy leads to "arbitrarily high [proxy] reward despite achieving no more [true reward]" than the base policy. This definition is quite similar to ours but does not capture that the true reward *decreases* when reward hacking occurs; furthermore, it is often impossible to obtain arbitrarily high proxy reward in practice. Finally, Fluri et al. (2024) define *error-regret* mismatch as when a proxy reward has low error on a training distribution compared to the true reward, but optimizing the proxy produces a suboptimal policy. This definition is also similar to ours but leaves the threshold for reward hacking ambiguous.

Furthermore, none of these works develop a general method for preventing reward hacking. In contrast, we will show that our definition leads to a natural regularization method for optimizing a proxy reward without allowing reward hacking. Also, unlike most of these works, we validate our definition and method through extensive experiments in realistic environments.

Figure 3: Unlike RLHF, which attempts to prevent reward hacking by regularizing *action distribution* (AD) divergence from the reference policy, our results suggest regularizing using *occupancy measure* (OM) divergence is more effective. These plots of the glucose monitoring environment show the typical ADs and OMs of two policies. $\pi$ is close to $\pi_{\text{ref}}$ in AD; it gives slightly less insulin. However, $\pi$'s optimization of the proxy also leads to a vastly different OM with typical glucose levels far outside the healthy range (dotted lines). Thus, regularizing ADs to be close to $\pi_{\text{ref}}$ is not enough to prevent reward hacking. Instead, divergence between the OMs better captures the reward hacking behavior.

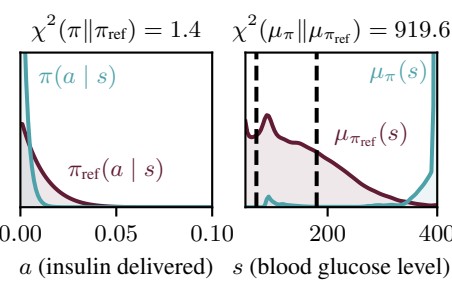

## 5 MITIGATING REWARD HACKING WITH OCCUPANCY MEASURE REGULARIZATION

We now discuss how our new definition of reward hacking motivates better methods for preventing it. Ideally, we would like to be able to optimize a proxy reward function and have it translate into an improvement in true reward over the reference policy. To understand how this might be possible, consider again the traffic environment. The reward hacking policy exhibits behavior which is very unlikely under the reference policy: human drivers hardly ever stop indefinitely on on-ramps. That is, optimizing the proxy reward leads to *out-of-distribution states and actions* where the correlation that made the proxy good in the first place breaks down. Visually, this can be seen in the second row of Figure 2: the reward hacking policy usually finds state-action pairs with high proxy reward and low true reward that were very unlikely to be reached by the reference policy.

Thus, one solution to prevent reward hacking could be to optimize the proxy reward while avoiding states that are unlikely under the reference policy. The following theorem formalizes this idea.

**Theorem 5.1.** *Suppose that $\tilde{R}$ is an $r$-correlated-proxy for the true reward function $R$, and let $\sigma_{\tilde{R}}$ and $\sigma_R$ be defined as in Definition 4.1. Then for any policy $\pi$ such that $\mu_\pi \ll \mu_{\pi_{\text{ref}}}$ (i.e., $\mu_{\pi_{\text{ref}}}(s,a) = 0 \Rightarrow \mu_\pi(s,a) = 0$), we have*

$$\frac{J(\pi, R) - J(\pi_{ref}, R)}{\sigma_R} \geq \frac{1}{r} \left( \frac{J(\pi, \tilde{R}) - J(\pi_{ref}, \tilde{R})}{\sigma_{\tilde{R}}} - \sqrt{(1-r^2)\chi^2\left(\mu_\pi \| \mu_{\pi_{ref}}\right)} \right), \qquad (3)$$

*where $\chi^2\left(\mu_\pi \| \mu_{\pi_{ref}}\right) = \mathbb{E}_{\mu_\pi}\left[\frac{\mu_\pi(s,a)}{\mu_{\pi_{ref}}(s,a)} - 1\right]$ is the $\chi^2$ divergence between $\mu_\pi$ and $\mu_{\pi_{ref}}$.*

See Appendix A for the proof. Equation (10) gives a lower bound on how much the policy $\pi$ improves over the reference policy $\pi_{\text{ref}}$ in terms of the true reward, normalized by the standard deviation of the true reward under the reference policy. Although this is exactly what we would like to maximize, we must instead optimize the right-hand side (RHS) of (10) since we cannot access the true reward directly. The RHS consists of two terms: one measuring the normalized improvement of the policy $\pi$'s proxy reward over the reference policy and the other penalizing the divergence of $\pi$'s occupancy measure from that of $\pi_{\text{ref}}$. Optimizing the first term alone often leads to reward hacking, but by both incentivizing $\pi$ to achieve high proxy reward and stay close to the reference policy, $\pi$ can achieve high true reward.

By scaling the RHS of (10) and removing terms that are constant with respect to $\pi$, we arrive at the following regularized policy optimization objective to avoid reward hacking:

$$\text{maximize} \quad J(\pi, \tilde{R}) - \lambda\sqrt{\chi^2\left(\mu_\pi \| \mu_{\pi_{\text{ref}}}\right)} \qquad \text{where} \qquad \lambda = \sigma_{\tilde{R}}\sqrt{1-r^2}. \qquad (4)$$

The amount of regularization needed to improve on the reference policy depends on the strength of correlation $r$. The higher the correlation, the lower the regularization strength $\sqrt{1-r^2}$. Given the prefactor of $\frac{1}{r}$ on the RHS of (10), it may appear that a lower correlation leads to a larger gain in true reward. However, Lemma A.2 shows that in fact, the lower bound decreases as a function of $r$.

While Theorem 5.1 does not *guarantee* that the true reward can be increased by optimizing (4), it does at least allow us to *provably avoid* reward hacking. In Theorem A.3 in the appendix, we show that it is difficult to guarantee an improvement in true reward in general. However, if the lower bound on

Figure 4: Our theory also suggests reward hacking can more effectively be prevented by regularizing with $\chi^2$ *divergence* instead of *KL divergence*. This plot illustrates how $\chi^2$ regularization is more effective at preventing reward hacking in RLHF. Both divergences can be written as the expectation of a function $g\big(\log\big(\mu_\pi(s,a)/\mu_{\pi_{\text{ref}}}(s,a)\big)\big)$ which increases the penalty on state-action pairs based on how far the log-ratio is from zero. The $g(\cdot)$ associated with KL divergence only increases slowly for large log-ratios, so policies trained with KL divergence may produce nonsensical text. In contrast, the $g(\cdot)$ for $\chi^2$ divergence increases exponentially, better constraining the LLM to produce text similar to the SFT policy.

the RHS of (10) can be increased above zero—which can be tested empirically—then we know that the true reward has also increased. In our experiments in Section 6, we show that in many realistic environments it is possible to increase the true reward by optimizing (4).

**Comparison to KL regularization in RLHF.**     Currently, RLHF calls for the application of a KL penalty between the distributions of responses generated by the optimized policy and the SFT policy (Stiennon et al., 2020; Bai et al., 2022). In our setting, we can write this as

$$\text{maximize} \quad J(\pi, \tilde{R}) - \lambda(1-\gamma)\mathbb{E}_\pi\left[\sum_{t=0}^\infty \gamma^t D_{\text{KL}}\big(\pi(\cdot \mid s_t) \parallel \mu_{\pi_{\text{ref}}}(\cdot \mid s_t)\big)\right]. \tag{5}$$

That is, in RLHF, the expected KL divergence between the action distributions of $\pi$ and $\pi_{\text{ref}}$ is penalized. Action distribution divergence is easy to calculate and optimize, and training LLMs with (5) seems to work well in practice. However, unlike our objective in (4), (5) lacks theoretical guarantees, and it is unclear if it works in other environments.

Our regularized objective in (4) differs from the regularization applied in RLHF in two key ways: our approach uses *occupancy measure* (OM) divergence instead of *action distribution* (AD) divergence, and it employs $\chi^2$ divergence instead of KL divergence. In this section, we provide intuition for why choosing OM over AD divergence and $\chi^2$ over KL divergence are more effective for preventing reward hacking. Then, in Section 6, we empirically explore applying different types of regularization to prevent reward hacking in the five environments we study.

**Occupancy measure vs. action distribution regularization.**     While AD regularization works well for RLHF, this may be because RLHF is essentially a contextual bandit problem, meaning that $\gamma = 0$; in this case, OM and AD divergence are equivalent (see Appendix A.3). However, in other cases, AD divergence may not suffice to prevent reward hacking behavior. This is because, in longer-horizon environments, a small change in action distribution at a single state can lead to a much higher probability of reaching undesirable states. Figure 3 shows an example of this in the glucose monitoring environment: policies that are close in action distribution regularization produce vastly different patient glucose levels. Occupancy measure regularization avoids this issue by directly preventing the distribution of glucose levels from differing too much from the reference policy. In Theorem A.5 in the appendix, we show that in general it is *impossible* to lower bound the improvement in true reward using almost any form of AD regularization, in contrast to our results on OM regularization.

$\chi^2$ **vs. KL divergence.**     Compared to KL divergence, $\chi^2$ divergence may be more effective for preventing reward hacking because it more strongly penalizes out-of-distribution state-action pairs. To illustrate this, we can write both divergences as expectations over functions of the log-ratio of the occupancy measures, which we denote $d(s,a)$:

$$D_{\text{KL}}(\mu_\pi \parallel \mu_{\pi_{\text{ref}}}) = \mathbb{E}_{\mu_\pi}\left[d(s,a) + e^{-d(s,a)}\right] \qquad \chi^2(\mu_\pi \parallel \mu_{\pi_{\text{ref}}}) = \mathbb{E}_{\mu_\pi}\left[e^{d(s,a)} + e^{-d(s,a)}\right]$$

$$\text{where} \quad d(s,a) = \log\big(\mu_\pi(s,a)/\mu_{\pi_{\text{ref}}}(s,a)\big). \tag{6}$$

| Method | Environment | | | | |
| | Traffic control ($\times 10^3$) | Pandemic mitigation | Glucose monitoring ($\times 10^3$) | RLHF | AI safety gridworld |
|---|---|---|---|---|---|
| Action dist. $\chi^2$ | $-1.29 \pm 0.10$ | $-12.29 \pm 0.05$ | $-74.8 \pm 11.8$ | $\mathbf{16.94} \pm 0.07$ | $6.24 \pm 0.09$ |
| State occupancy $\chi^2$ | $-2.18 \pm 0.38$ | $-10.68 \pm 0.15$ | $-54.7 \pm 1.0$ | — | $9.07 \pm 0.06$ |
| State-action occupancy $\chi^2$ | $\mathbf{-1.15} \pm 0.05$ | $-11.17 \pm 0.17$ | $\mathbf{-47.6} \pm 0.6$ | — | $\mathbf{9.17} \pm 0.11$ |
| Action dist. KL | $-1.33 \pm 0.05$ | $-12.20 \pm 0.06$ | $-73.4 \pm 8.3$ | $16.81 \pm 0.27$ | $6.33 \pm 0.11$ |
| State occupancy KL | $-1.34 \pm 22.6$ | $\mathbf{-10.24} \pm 0.54$ | $-58.4 \pm 3.4$ | — | $7.07 \pm 0.11$ |
| State-action occupancy KL | $-1.25 \pm 0.06$ | $-11.73 \pm 0.19$ | $-48.9 \pm 0.5$ | — | $6.86 \pm 0.17$ |
| $\pi_{\text{ref}}$ | $-2.28 \pm 0.00$ | $-12.26 \pm 0.00$ | $-72.6 \pm 0.0$ | $16.37 \pm 0.00$ | $5.86 \pm 0.00$ |
| No regularization | $-57.38 \pm 3.53$ | $-29.57 \pm 6.86$ | $-599.0 \pm 1.6$ | $9.16 \pm 0.80$ | $2.35 \pm 0.14$ |
| Training with true reward | $-0.93 \pm 0.11$ | $-2.65 \pm 0.83$ | $-43.4 \pm 0.8$ | — | $8.54 \pm 0.12$ |

Table 1: We compare using various types of regularization to prevent reward hacking in the five environments form Figure 2. The median true reward and standard deviation across 5 random seeds is shown for the best regularization coefficient for each type of regularization. The bottom rows show results for the baselines: the reference policy $\pi_{\text{ref}}$, a policy trained on the proxy reward without regularization (exhibiting reward hacking), and a policy trained on the true reward function (impossible in practice, but included as an upper bound on performance). We find that occupancy measure regularization consistently improves on action distribution regularization, and that $\chi^2$ divergence often outperforms KL divergence.

As $d(s, a)$ increases, the optimized policy is visiting state-action pairs that are less likely under the reference policy. However, KL divergence only penalizes $d(s, a)$ linearly, while $\chi^2$ penalizes it exponentially, resulting in stronger regularization even with a low coefficient. Figure 4 plots the functions in (6) and shows how in practice $\chi^2$ divergence better prevents reward hacking in RLHF.

## 6 EXPERIMENTS

We now show that our theoretical results—which suggest $\chi^2$ occupancy measure regularization can prevent reward hacking—translate to empirical success in realistic environments.

**Occupancy measure regularization in practice.** Occupancy measure regularization is more difficult to implement in practice compared to action distribution regularization. While AD regularization can be added as a loss term to deep RL algorithms like proximal policy optimization (PPO) (Schulman et al., 2017), OM divergences cannot be calculated in closed form. Instead, we follow several previous works (e.g., Ho & Ermon 2016; Kang et al. 2018) and use a discriminator network to approximate OM divergences. Specifically, in Appendix B, we show the objective in (4) can be optimized via policy gradient with an adjusted reward function that depends on a discriminator $\hat{d}_\phi$:

$$R'(s, a) = \tilde{R}(s, a) - \frac{\lambda}{\sqrt{\widehat{\chi^2}}} e^{\hat{d}_\phi(s,a)} \qquad \text{where} \qquad \widehat{\chi^2} = \mathbb{E}_{\mu_\pi}\left[ e^{\hat{d}_\phi(s_t, a_t)} - 1 \right]$$

$$\text{and} \qquad \phi = \arg\min_{\phi'} \; \mathbb{E}_{\mu_\pi}\left[ \log(1 + e^{-\hat{d}_{\phi'}(s,a)}) \right] + \mathbb{E}_{\mu_{\pi_{\text{ref}}}}\left[ \log(1 + e^{\hat{d}_{\phi'}(s,a)}) \right]. \qquad (7)$$

That is, optimizing a discriminator network $\hat{d}_\phi$ to minimize the given loss can be used to estimate and optimize $\chi^2$ OM divergence. We alternately train $\hat{d}_\phi$ via gradient descent on (7) and the policy $\pi$ via PPO based on the adjusted reward. We call this algorithm Occupancy-Regularized Policy Optimization (ORPO). See Appendix B for a full derivation of these approximations and Algorithm 1 for a formal description; we also describe how to regularize based on OM KL divergence.

**Experimental setup.** In each of the five environments shown in Figure 2, we train policies with four types of regularization towards the reference policy: AD KL, AD $\chi^2$, OM KL, and OM $\chi^2$. In Appendix A.5, we show that Theorem 5.1 also holds for state-only occupancy measures if the environment's reward function does not depend on the action; thus, we experiment with regularizing based on both state-action and state-only OM divergence. For each environment and type of regularization, we test a number of regularization coefficients $\lambda$. Theorem 5.1 suggests setting $\lambda = \sigma_{\tilde{R}}\sqrt{1 - r^2}$ for an $r$-correlated proxy, so for $\chi^2$ regularization we test a range of values $\lambda = c\,\sigma_{\tilde{R}}$ from $c = 1$ to $10^{-2}$ ($10^{-4}$ for RLHF). Since it is less clear theoretically how to set the coefficient $\lambda$ for KL regularization, we experiment with a wider range of values.

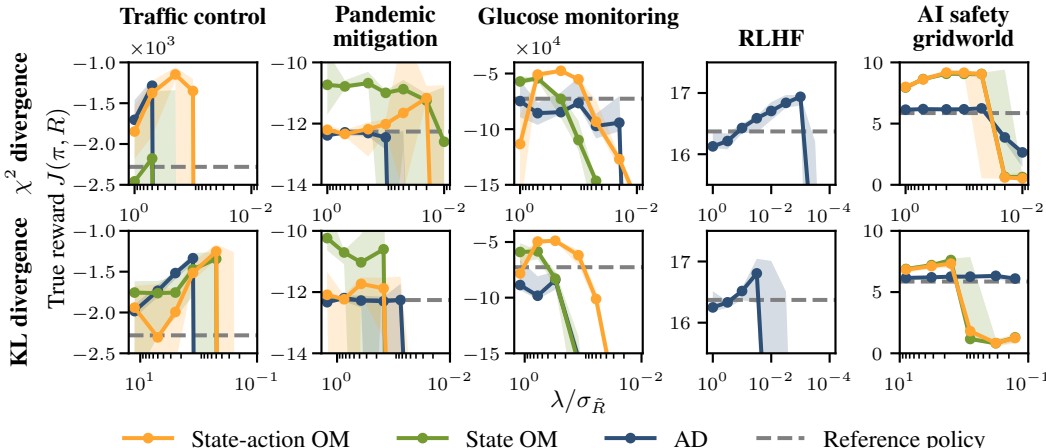

Figure 5: The true reward achieved by policies regularized with varying amounts of action distribution or occupancy measure regularization using $\chi^2$ and KL divergence. The x-axis is the regularization coefficient $\lambda$ normalized by the standard deviation of proxy rewards under the reference policy. Dots indicate the median reward and the shaded area is the range over random seeds. For RLHF, AD and OM regularization are equivalent, which is why OM regularization results are not shown for that column.

We train each combination of $\{\chi^2$ divergence, KL divergence$\} \times \{$AD, state OM, state-action OM$\}$ $\times \{\lambda_1, \lambda_2, \dots\}$ with five random seeds and measure the resulting policies' expected returns under the true reward. As a baseline, we train a policy without any regularization, which leads to reward hacking in all environments. We also train a policy directly on the true reward function as an upper limit for performance. In RLHF, we only consider AD regularization since it is equivalent to OM regularization for LLM chatbots (see Appendix A.3). We do not train a policy on the true reward for RLHF as we found it could be hacked with enough optimization pressure. See Appendix D for all hyperparameter and experiment details.

**Results.** The results of our experiments are shown in Table 1 and Figure 5. Table 1 shows the median true reward with the best coefficient for each type of regularization. We find that OM regularization consistently outperforms AD regularization across the four non-RLHF environments. In two environments (glucose monitoring and pandemic mitigation), AD regularization fails to improve on the reference policy's true reward at all. Furthermore, $\chi^2$ regularization tends to perform similarly to or better than KL regularization across all environments. In RLHF in particular, $\chi^2$ regularization leads to a larger improvement over the reference policy compared to the industry-standard KL penalty, and is more stable across seeds as well.

In Figure 5, we show the true reward achieved when training with each type of regularization across a range of $\lambda$ values. In addition to performing best with an optimal coefficient, $\chi^2$ and OM regularization seem to also perform well over a larger range of coefficients compared to AD regularization. In Appendix E, we present the full results of our experiments and ablations of ORPO.

## 7  CONCLUSION

We have introduced a new definition for reward hacking based on the correlation between a proxy reward function and the unknown true reward that breaks down when optimizing the proxy. Furthermore, we leveraged this definition to show theoretically and empirically that $\chi^2$ occupancy measure regularization can more effectively prevent reward hacking than current approaches. Our results have implications for settings, like RLHF, where RL is used to optimize complex, hard-to-specify objectives. We suggest that the heuristic KL penalty used currently should be replaced by a more principled form of regularization. While OM and AD regularization are equivalent for today's formulation of RLHF as a contextual bandit, they will likely diverge in the near future when LLM-based agents are optimized over multi-turn conversations or with tool use (Wang et al., 2023b; Abdulhai et al., 2023; Shani et al., 2024). Thus, our results provide a principled path to continuing to ensure the safety of increasingly powerful AI systems.

ACKNOWLEDGMENTS

We would like to thank Cam Allen, Micah Carroll, Dibya Ghosh, Katie Kang, and Sam Toyer for helpful discussions and feedback on drafts.

This work was supported by the Semiconductor Research Corporation (SRC)'s Center for the Co-Design of Cognitive Systems (COCOSYS) as well as the NSF's Human Centered Computing (HCC) program. Cassidy Laidlaw is supported by an Open Philanthropy AI Fellowship and a National Defense Science and Engineering Graduate (NDSEG) Fellowship.

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

# Appendix

## A  PROOFS AND ADDITIONAL THEORETICAL RESULTS

### A.1  PROOF OF THEOREM 5.1

**Theorem 5.1.** *Suppose that $\tilde{R}$ is an $r$-correlated-proxy for the true reward function $R$, and let $\sigma_{\tilde{R}}$ and $\sigma_R$ be defined as in Definition 4.1. Then for any policy $\pi$ such that $\mu_\pi \ll \mu_{\pi_{ref}}$ (i.e., $\mu_{\pi_{ref}}(s,a) = 0 \Rightarrow \mu_\pi(s,a) = 0$), we have*

$$\frac{J(\pi, R) - J(\pi_{ref}, R)}{\sigma_R} \geq \frac{1}{r} \left( \frac{J(\pi, \tilde{R}) - J(\pi_{ref}, \tilde{R})}{\sigma_{\tilde{R}}} - \sqrt{(1-r^2)\chi^2\left(\mu_\pi \| \mu_{\pi_{ref}}\right)} \right), \qquad (3)$$

*where $\chi^2\left(\mu_\pi \| \mu_{\pi_{ref}}\right) = \mathbb{E}_{\mu_\pi}\left[ \frac{\mu_\pi(s,a)}{\mu_{\pi_{ref}}(s,a)} - 1 \right]$ is the $\chi^2$ divergence between $\mu_\pi$ and $\mu_{\pi_{ref}}$.*

*Proof.* For simplicity of exposition, define

$$Z(s,a) = \frac{R(s,a) - J(\pi_{\text{ref}}, R)}{\sigma_R} \qquad \text{and} \qquad \tilde{Z}(s,a) = \frac{\tilde{R}(s,a) - J(\pi_{\text{ref}}, \tilde{R})}{\sigma_{\tilde{R}}}.$$

Combining (1) and (2), we can see that the return of a policy can be re-written as the expected reward for states and actions sampled from the occupancy measure:

$$J(\pi, R) = \sum_{s,a \in \mathcal{S} \times \mathcal{A}} \mu_\pi(s,a) R(s,a) = \mathbb{E}_{\mu_\pi}[R(s,a)]. \qquad (8)$$

Using (8), we can rewrite (10) as

$$\mathbb{E}_{\mu_\pi}\left[ \tilde{Z}(s,a) - r\, Z(s,a) \right] \leq \sqrt{(1-r^2)\chi^2\left(\mu_\pi \| \mu_{\pi_{\text{ref}}}\right)}.$$

Then, the left hand side can be rewritten as

$$\mathbb{E}_{\mu_\pi}\left[ \tilde{Z}(s,a) - r\, Z(s,a) \right]$$
$$= \mathbb{E}_{\mu_\pi}\left[ \tilde{Z}(s,a) - r\, Z(s,a) \right] - \mathbb{E}_{\mu_{\pi_{\text{ref}}}}\left[ \tilde{Z}(s,a) - r\, Z(s,a) \right] + \mathbb{E}_{\mu_{\pi_{\text{ref}}}}\left[ \tilde{Z}(s,a) - r\, Z(s,a) \right]$$
$$= \mathbb{E}_{\mu_\pi}\left[ \tilde{Z}(s,a) - r\, Z(s,a) \right] - \mathbb{E}_{\mu_{\pi_{\text{ref}}}}\left[ \tilde{Z}(s,a) - r\, Z(s,a) \right],$$

since by definition $\mathbb{E}_{\mu_{\pi_{\text{ref}}}}[\tilde{Z}(s,a)] = \mathbb{E}_{\mu_{\pi_{\text{ref}}}}[Z(s,a)] = 0$. Applying the Cauchy-Schwartz inequality to this difference gives

$$\mathbb{E}_{\mu_\pi}\left[ \tilde{Z}(s,a) - r\, Z(s,a) \right] - \mathbb{E}_{\mu_{\pi_{\text{ref}}}}\left[ \tilde{Z}(s,a) - r\, Z(s,a) \right]$$
$$= \sum_{(s,a) \in \mathcal{S} \times \mathcal{A}} \left[ \tilde{Z}(s,a) - r\, Z(s,a) \right] \left[ \mu_\pi(s,a) - \mu_{\pi_{\text{ref}}}(s,a) \right]$$
$$= \sum_{(s,a) \in \mathcal{S} \times \mathcal{A}} \left[ \sqrt{\mu_{\pi_{\text{ref}}}(s,a)} \left( \tilde{Z}(s,a) - r\, Z(s,a) \right) \right] \left[ \frac{\mu_\pi(s,a) - \mu_{\pi_{\text{ref}}}(s,a)}{\sqrt{\mu_{\pi_{\text{ref}}}(s,a)}} \right]$$
$$\leq \sqrt{\left( \sum_{(s,a) \in \mathcal{S} \times \mathcal{A}} \mu_{\pi_{\text{ref}}}(s,a) \left[ \tilde{Z}(s,a) - r\, Z(s,a) \right]^2 \right) \left( \sum_{(s,a) \in \mathcal{S} \times \mathcal{A}} \frac{(\mu_\pi(s,a) - \mu_{\pi_{\text{ref}}}(s,a))^2}{\mu_{\pi_{\text{ref}}}(s,a)} \right)}$$
$$= \sqrt{\mathbb{E}_{\mu_{\pi_{\text{ref}}}}\left[ \left( \tilde{Z}(s,a) - r\, Z(s,a) \right)^2 \right] \chi^2\left(\mu_\pi \| \mu_{\pi_{\text{ref}}}\right)}. \qquad (9)$$

The expectation can be calculated as

$$
\mathbb{E}_{\mu_{\pi_{\mathrm{ref}}}}\left[\left(\tilde{Z}(s,a) - r\,Z(s,a)\right)^2\right]
$$

$$
= \mathrm{Var}_{\mu_{\pi_{\mathrm{ref}}}}\left(\tilde{Z}(s,a) - r\,Z(s,a)\right)
$$

$$
= \mathrm{Var}_{\mu_{\pi_{\mathrm{ref}}}}\left(\tilde{Z}(s,a)\right) + r^2\,\mathrm{Var}_{\mu_{\pi_{\mathrm{ref}}}}\left(Z(s,a)\right) - 2r\,\mathrm{Cov}_{\mu_{\pi_{\mathrm{ref}}}}\left(\tilde{Z}(s,a), Z(s,a)\right)
$$

$$
= 1 - r^2,
$$

using the fact that both reward functions have unit variance under the reference policy and that their correlation is $r$. Plugging this into (9) gives the desired result. $\qquad\square$

### A.1.1 NEAR-OPTIMAL REFERENCE POLICIES

While Theorem 5.1 shows that optimizing the proxy reward with regularization can improve on the reference policy's reward, it does not guarantee that the learned policy will be near-optimal. However, a simple corollary shows that if the reference policy is near-optimal, then the learned policy will also be near-optimal.

**Corollary A.1.** *Suppose that $\tilde{R}$ is an $r$-correlated-proxy for the true reward function $R$, and let $\sigma_{\tilde{R}}$ and $\sigma_R$ be defined as in Definition 4.1. Furthermore, suppose that the reference policy $\pi_{ref}$ is near-optimal:*

$$
J(\pi_{ref}, R) \geq \max_{\pi^*} J(\pi^*, R) - \epsilon\sigma_R,
$$

*for some $\epsilon > 0$.*

*Then for any policy $\pi$ such that $\mu_\pi \ll \mu_{\pi_{ref}}$, we have*

$$
\frac{\max_{\pi^*} J(\pi^*, R) - J(\pi, R)}{\sigma_R} \leq \epsilon - \frac{1}{r}\left(\frac{J(\pi, \tilde{R}) - J(\pi_{ref}, \tilde{R})}{\sigma_{\tilde{R}}} - \sqrt{(1-r^2)\chi^2\left(\mu_\pi \| \mu_{\pi_{ref}}\right)}\right). \tag{10}
$$

This result bounds the suboptimality gap—how close the learned policy is to optimal—in terms of the suboptimality gap of the reference policy and the increase in the regularized proxy reward. The proof is straightforward and follows from Theorem 5.1.

### A.1.2 UNDERSTANDING THE LOWER BOUND IN THEOREM 5.1

Denote by

$$
L(\pi) = \frac{1}{r}\left(\frac{J(\pi, \tilde{R}) - J(\pi_{\mathrm{ref}}, \tilde{R})}{\sigma_{\tilde{R}}} - \sqrt{(1-r^2)\chi^2\left(\mu_\pi \| \mu_{\pi_{\mathrm{ref}}}\right)}\right) \tag{11}
$$

the lower bound on increase in the true reward which is the RHS of (10). One surprising observation is that this lower bound seems to be *increasing* as the proxy becomes *less* correlated with the true reward. This would suggest that a less correlated proxy leads to better optimization of the true reward. However, as the following lemma shows, $L(\pi)$ is actually decreasing in $r$.

**Lemma A.2.** *Under the same conditions as Theorem 5.1, the lower bound $L(\pi)$ satisfies*

$$
L(\pi) \leq \frac{1 - \sqrt{1-r^2}}{r}\sqrt{\chi^2\left(\mu_\pi \| \mu_{\pi_{ref}}\right)}. \tag{12}
$$

This shows that the lower bound can be at most a factor of $\frac{1-\sqrt{1-r^2}}{r}$ times the divergence between the learned and reference policies' occupancy measures. This factor is increasing in $r$ and asymptotes to $r/2$ as $r \to 0$.

*Proof.* Using the same notation as the proof of Theorem 5.1, we can rewrite the lower bound as

$$L(\pi) = \frac{1}{r} \left( \mathbb{E}_\pi \left[ \tilde{Z}(s,a) \right] - \sqrt{(1-r^2)\chi^2 \left( \mu_\pi \| \mu_{\pi_{\mathrm{ref}}} \right)} \right).$$

Following a similar argument to that in the proof of Theorem 5.1, we can write

$$\mathbb{E}_\pi \left[ \tilde{Z}(s,a) \right] = \mathbb{E}_\pi \left[ \tilde{Z}(s,a) \right] - \mathbb{E}_{\pi_{\mathrm{ref}}} \left[ \tilde{Z}(s,a) \right]$$

$$= \sum_{(s,a)\in\mathcal{S}\times\mathcal{A}} \left[ \sqrt{\mu_{\pi_{\mathrm{ref}}}(s,a)} \tilde{Z}(s,a) \right] \left[ \frac{\mu_\pi(s,a) - \mu_{\pi_{\mathrm{ref}}}(s,a)}{\sqrt{\mu_{\pi_{\mathrm{ref}}}(s,a)}} \right]$$

$$\leq \sqrt{ \left( \sum_{(s,a)\in\mathcal{S}\times\mathcal{A}} \mu_{\pi_{\mathrm{ref}}}(s,a)\tilde{Z}(s,a)^2 \right) \left( \sum_{(s,a)\in\mathcal{S}\times\mathcal{A}} \frac{(\mu_\pi(s,a) - \mu_{\pi_{\mathrm{ref}}}(s,a))^2}{\mu_{\pi_{\mathrm{ref}}}(s,a)} \right) }$$

$$= \sqrt{ \mathbb{E}_{\mu_{\pi_{\mathrm{ref}}}} \left[ \tilde{Z}(s,a)^2 \right] \chi^2 \left( \mu_\pi \| \mu_{\pi_{\mathrm{ref}}} \right) }$$

$$= \sqrt{ \chi^2 \left( \mu_\pi \| \mu_{\pi_{\mathrm{ref}}} \right) }.$$

Combining this with the definition of $L(\pi)$ gives

$$L(\pi) \leq \frac{1}{r} \left( \sqrt{\chi^2 \left( \mu_\pi \| \mu_{\pi_{\mathrm{ref}}} \right)} - \sqrt{(1-r^2)\chi^2 \left( \mu_\pi \| \mu_{\pi_{\mathrm{ref}}} \right)} \right)$$

$$= \frac{1 - \sqrt{1-r^2}}{r} \sqrt{\chi^2 \left( \mu_\pi \| \mu_{\pi_{\mathrm{ref}}} \right)},$$

which completes the proof. $\square$

### A.1.3 Is the Lower Bound Optimizable?

While Theorem 5.1 shows that the increase in true reward over the reference policy can be lower-bounded by the the proxy reward minus a regularization term, it is not clear if it is actually possible to increase the lower bound $L(\pi)$ as defined in (11). For example, if the reference policy is already optimal with respect to the proxy reward, then clearly $L(\pi) \leq 0$. As another example, suppose it possible to improve $\pi_{\mathrm{ref}}$ with respect to both the true and proxy rewards, but only by visiting a state-action pair never visited by $\pi_{\mathrm{ref}}$. In this case, it is also impossible to improve the lower bound in Theorem 5.1 while obeying the requirement that $\mu_\pi \ll \mu_{\pi_{\mathrm{ref}}}$.

We prove two results relating to whether $L(\pi)$ can be increased above zero. Lemma A.3 shows that in general it is difficult to show when the lower bound can be optimized—there are general counterexamples where it cannot be positive. However, Lemma A.4 shows that there *are* MDPs with $r$-correlated proxies for any $r$ where the lower bound can be positive.

Furthermore, as we show in our experiments, in many realistic environments it does appear that the lower bound is optimizable. Furthermore, Theorem 5.1 still allows for safe optimization of the proxy reward: even if the it not possible to increase the lower bound above zero, optimizing $L(\pi)$ will at least prevent reward hacking.

**Lemma A.3.** *Fix any $r \in (0,1)$. Then there is an MDP with a true reward function $R$, a proxy reward $\tilde{R}$, and a reference policy $\pi_{ref}$ such that $\tilde{R}$ is an $r$-correlated proxy that can be improved upon in both true and proxy reward by a policy $\pi^*$:*

$$J(\pi^*, R) > J(\pi_{ref}, R)$$
$$J(\pi^*, \tilde{R}) > J(\pi_{ref}, \tilde{R})$$
$$\mu_{\pi^*} \ll \mu_{\pi_{ref}}.$$

*However, for any policy $\pi$ such that $\mu_\pi \ll \mu_{\pi_{ref}}$,*

$$L(\pi) = \frac{1}{r}\left(\frac{J(\pi,\tilde{R}) - J(\pi_{ref},\tilde{R})}{\sigma_{\tilde{R}}} - \sqrt{(1-r^2)\chi^2\left(\mu_\pi\|\mu_{\pi_{ref}}\right)}\right) \le 0.$$

That is, Lemma A.3 shows that there is an MDP where there exists a policy $\pi^*$ that improves on the reference policy in both true reward and proxy reward, but it is still not possible to increase the lower bound $L(\pi)$ above zero. This suggests that it is difficult to specify general conditions under which $L(\pi)$ can exceed zero. Thus, we rely on the empirical evidence from our experiments to show that the lower bound is often optimizable.

*Proof.* We consider MDPs with discount $\gamma = 0$, such that the transition probabilities are not relevant; only the initial state distribution $\mu_0$ and the reward functions specify the MDP. We split the analysis into two cases depending on whether $r \le 1/2$ or $r \ge 1/2$.

**Case 1: $r \le 1/2$.** We define an MDP with two states $s_1, s_2$ and two actions $a_1, a_2$, with initial state distribution and rewards as follows:

$$\mu_0(s_1) = \frac{1}{1+r} \qquad R(s_1,a_1) = \sqrt{\frac{r}{1-r}} \qquad \tilde{R}(s_1,a_1) = \sqrt{\frac{r}{1-r}}$$

$$R(s_1,a_2) = -\sqrt{\frac{1-r}{r}} \qquad \tilde{R}(s_1,a_2) = 0$$

$$\mu_0(s_2) = \frac{r}{1+r} \qquad R(s_2,\cdot) = 0 \qquad \tilde{R}(s_2,\cdot) = -\sqrt{\frac{1-r}{r}}$$

Furthermore, we define $\pi_{ref}(a_1 \mid s_1) = 1 - r$ and $\pi_{ref}(a_2 \mid s_1) = r$, and $\pi_{ref}(a_1 \mid s_2) = 1$. Based on this, simple algebra shows the following facts:

$$\mu_{\pi_{ref}}(s_1,a_1) = \frac{1-r}{1+r} \qquad \mu_{\pi_{ref}}(s_1,a_2) = \frac{r}{1+r}$$

$$J(\pi_{ref},R) = J(\pi_{ref},\tilde{R}) = 0 \qquad \sigma_R = \sigma_{\tilde{R}} = \sqrt{\frac{1}{1+r}}$$

$$\mathbb{E}_{\mu_{\pi_{ref}}}\left[R(s,a)\tilde{R}(s,a)\right] = \frac{r}{1+r}.$$

Based on this, it is clear that $\tilde{R}$ is an $r$-correlated proxy. Furthermore, letting $\pi^*(a_1 \mid s_1) = 1$ and $\pi^*(a_1 \mid s_2) = 1$, we have

$$J(\pi^*, R) = \frac{1}{1+r}\sqrt{\frac{r}{1-r}} > 0 = J(\pi_{ref}, R)$$

$$J(\pi^*, \tilde{R}) = \frac{r}{1+r}\sqrt{\frac{r}{1-r}} > 0 = J(\pi_{ref}, \tilde{R})$$

$$\mu_{\pi^*} \ll \mu_{\pi_{ref}},$$

satisfying the conditions in the lemma.

Now, consider any policy $\pi$ such that $\mu_\pi \ll \mu_{\pi_{ref}}$. We can calculate the lower bound $L(\pi)$ (ignoring the prefactor of $\frac{1}{r}$) as

$$J(\pi,\tilde{R})\sqrt{1+r} - \sqrt{(1-r^2)\chi^2\left(\mu_\pi\|\mu_{\pi_{ref}}\right)}. \tag{13}$$

Since $\mu_\pi \ll \mu_{\pi_{ref}}$, $\pi(a_1 \mid s_2) = 1$, so it can only differ from $\pi_{ref}$ in state $s_1$. Let $\delta = \pi(a_1 \mid s_1) - \pi_{ref}(a_1 \mid s_1)$. Then, we can write the first term of (13) as

$$J(\pi,\tilde{R})\sqrt{1+r} = \delta\frac{1}{1+r}\sqrt{\frac{r}{1-r}}\sqrt{1+r} = \delta\sqrt{\frac{r}{1-r^2}}.$$

Note that the $\chi^2$ divergence between distributions $\mu$ and $\nu$ can be alternatively written as

$$\chi^2(\mu\|\nu) = \sum_{s,a} \frac{(\mu(s,a) - \nu(s,a))^2}{\nu(s,a)}.$$

Therefore, the $\chi^2$ divergence between $\mu_\pi$ and $\mu_{\pi_{\text{ref}}}$ can be lower bounded as

$$\chi^2\left(\mu_\pi\|\mu_{\pi_{\text{ref}}}\right) \geq \frac{(\mu_\pi(s_1,a_1) - \mu_{\pi_{\text{ref}}}(s_1,a_1))^2}{\mu_{\pi_{\text{ref}}}(s_1,a_1)} = \frac{\left(\frac{1-r+\delta}{1+r} - \frac{1-r}{1+r}\right)^2}{\frac{1-r}{1+r}} = \frac{\delta^2}{1-r^2}.$$

This leads to the bound on (13):

$$rL(\pi) \leq \delta\sqrt{\frac{r}{1-r^2}} - \sqrt{(1-r^2)\frac{\delta^2}{1-r^2}} = \delta\sqrt{\frac{r}{1-r^2}} - |\delta|.$$

Clearly if $\delta \leq 0$ this is non-positive, and if $\delta > 0$ it is also non-positive since $\sqrt{r/(1-r^2)} < 1$ as long as $r \leq 1/2$. Thus, the lower bound cannot be increased above zero in this case.

**Case 2: $r \geq 1/2$.** In this case, we define an MDP with three states $s_1, s_2, s_3$ and two actions $a_1, a_2$, with initial state distribution and rewards as follows:

$$\mu_0(s_1) = \frac{2r^2 - 2r + 1}{r^2 - r + 1} \qquad R(s_1, a_1) = \sqrt{\frac{1-r}{r}} \qquad \tilde{R}(s_1, a_1) = \sqrt{\frac{1-r}{r}}$$

$$R(s_1, a_2) = -\sqrt{\frac{r}{1-r}} \qquad \tilde{R}(s_1, a_2) = 0$$

$$\mu_0(s_2) = \frac{(1-r)^2}{r^2 - r + 1} \qquad R(s_2, \cdot) = 0 \qquad \tilde{R}(s_2, \cdot) = -\sqrt{\frac{r}{1-r}}$$

$$\mu_0(s_3) = \frac{(1-r)(2r-1)}{r^2 - r + 1} \qquad R(s_3, \cdot) = -\sqrt{\frac{r}{1-r}} \qquad \tilde{R}(s_3, \cdot) = -\sqrt{\frac{r}{1-r}}$$

We define the reference policy $\pi_{\text{ref}}$ as follows:

$$\pi_{\text{ref}}(a_1 \mid s_1) = \frac{r^2}{2r^2 - 2r + 1} \qquad\qquad \pi_{\text{ref}}(a_2 \mid s_1) = \frac{(1-r)^2}{2r^2 - 2r + 1}$$

$$\pi_{\text{ref}}(a_1 \mid s_2) = 1 \qquad\qquad \pi_{\text{ref}}(a_1 \mid s_3) = 1.$$

As above, we can show the following facts:

$$\mu_{\pi_{\text{ref}}}(s_1, a_1) = \frac{r^2}{r^2 - r + 1} \qquad\qquad \mu_{\pi_{\text{ref}}}(s_1, a_2) = \frac{(1-r)^2}{r^2 - r + 1}$$

$$J(\pi_{\text{ref}}, R) = J(\pi_{\text{ref}}, \tilde{R}) = 0 \qquad\qquad \sigma_R = \sigma_{\tilde{R}} = \sqrt{\frac{r}{r^2 - r + 1}}$$

$$\mathbb{E}_{\mu_{\pi_{\text{ref}}}}\left[R(s,a)\tilde{R}(s,a)\right] = \frac{r^2}{r^2 - r + 1}.$$

Again, this shows that $\tilde{R}$ is an $r$-correlated proxy. Letting $\pi^*(a_1 \mid s_1) = \pi^*(a_1 \mid s_2) = \pi^*(a_1 \mid s_3) = 1$, we have

$$J(\pi^*, R) = \frac{1-r}{r^2 - r + 1}\sqrt{\frac{1-r}{r}} > 0 = J(\pi_{\text{ref}}, R)$$

$$J(\pi^*, \tilde{R}) = \frac{(1-r)^2}{r^2 - r + 1}\sqrt{\frac{1-r}{r}} > 0 = J(\pi_{\text{ref}}, \tilde{R})$$

$$\mu_{\pi^*} \ll \mu_{\pi_{\text{ref}}},$$

satisfying the conditions in the lemma.

Now, consider any policy $\pi$ such that $\mu_\pi \ll \mu_{\pi_{\text{ref}}}$. We can calculate the lower bound $L(\pi)$ (again ignoring the prefactor of $\frac{1}{r}$) as

$$J(\pi, \tilde{R})\sqrt{\frac{r^2 - r + 1}{r}} - \sqrt{(1 - r^2)\chi^2\left(\mu_\pi \| \mu_{\pi_{\text{ref}}}\right)}. \tag{14}$$

As in the previous case, let $\delta = \pi(a_1 \mid s_1) - \pi_{\text{ref}}(a_1 \mid s_1)$. Then, we can write the first term of (14) as

$$J(\pi, \tilde{R})\sqrt{\frac{r^2 - r + 1}{r}} = \delta\frac{2r^2 - 2r + 1}{r^2 - r + 1}\sqrt{\frac{1 - r}{r}}\sqrt{\frac{r^2 - r + 1}{r}} = \delta\frac{2r^2 - 2r + 1}{r}\sqrt{\frac{1 - r}{r^2 - r + 1}}.$$

The $\chi^2$ divergence between $\mu_\pi$ and $\mu_{\pi_{\text{ref}}}$ can be lower bounded as

$$\chi^2\left(\mu_\pi \| \mu_{\pi_{\text{ref}}}\right) \geq \frac{(\mu_\pi(s_1, a_1) - \mu_{\pi_{\text{ref}}}(s_1, a_1))^2}{\mu_{\pi_{\text{ref}}}(s_1, a_1)} = \frac{\left(\frac{2r^2 - 2r + 1}{r^2 - r + 1}\delta\right)^2}{\frac{r^2}{r^2 - r + 1}} = \delta^2\left(\frac{2r^2 - 2r + 1}{r}\right)^2\frac{1}{r^2 - r + 1}.$$

This leads to the bound on (14):

$$rL(\pi) \leq \delta\frac{2r^2 - 2r + 1}{r}\sqrt{\frac{1 - r}{r^2 - r + 1}} - \sqrt{(1 - r^2)\delta^2\left(\frac{2r^2 - 2r + 1}{r}\right)^2\frac{1}{r^2 - r + 1}}$$

$$= \frac{2r^2 - 2r + 1}{r\sqrt{r^2 - r + 1}}\left(\delta\sqrt{1 - r} - |\delta|\sqrt{1 - r^2}\right).$$

The prefactor is positive, and as in the previous case, if $\delta \leq 0$ then the bound in non-positive. If $\delta > 0$, then the bound is also non-positive since $\sqrt{1 - r} < \sqrt{1 - r^2}$ for any $r \in (0, 1)$. Thus, the lower bound cannot be increased above zero in this case either, completing the proof. $\qquad\square$

**Lemma A.4.** *Fix any $r \in (0, 1)$. Then there is an MDP with a true reward function $R$, a proxy reward $\tilde{R}$, and a reference policy $\pi_{\text{ref}}$ such that $\tilde{R}$ is an $r$-correlated proxy and:*

1. *There is a policy $\pi^*$ such that $L(\pi^*) > 0$.*

2. *Any optimal policy with respect to $L(\cdot)$ is also an optimal policy with respect to the true reward function:*

$$\arg\max_\pi L(\pi) \subseteq \arg\max_\pi J(\pi, R).$$

Lemma A.4 shows that no matter how low the correlation coefficient $r$ is between the true and proxy rewards, there is at least *some* MDP where the lower bound $L(\pi)$ can be positive. Furthermore, in this MDP, maximizing $L(\pi)$ actually leads to an optimal policy with respect to the true reward function.

*Proof.* As in the proof of Lemma A.3, we construct an MDP with discount factor $\gamma = 0$ that has three states and two actions. The initial state distribution and reward functions are given by

$$\mu_0(s_1) = \frac{1 - r}{4} \qquad \mu_0(s_2) = \frac{3 + r}{8} \qquad \mu_0(s_3) = \frac{3 + r}{8}$$

$$R(s_1, a_1) = \tilde{R}(s_1, a_1) = \sqrt{2\frac{1 + r}{1 - r}} \qquad R(s_2, \cdot) = \sqrt{2\frac{1 - r}{3 + r}} \qquad R(s_3, \cdot) = -\sqrt{2\frac{1 - r}{3 + r}}$$

$$R(s_1, a_2) = R(s_1, a_2) = -\sqrt{2\frac{1 - r}{1 + r}} \qquad \tilde{R}(s_2, \cdot) = -\sqrt{2\frac{1 - r}{3 + r}} \qquad \tilde{R}(s_3, \cdot) = \sqrt{2\frac{1 - r}{3 + r}}.$$

We define the reference policy $\pi_{\text{ref}}$ as follows:

$$\pi_{\text{ref}}(a_1 \mid s_1) = \pi_{\text{ref}}(a_2 \mid s_1) = 1/2 \qquad \pi_{\text{ref}}(a_1 \mid s_2) = 1 \qquad \pi_{\text{ref}}(a_1 \mid s_3) = 1.$$

We can show the following facts:

$$J(\pi_{\text{ref}}, R) = J(\pi_{\text{ref}}, \tilde{R})$$

$$= \frac{1-r}{8}\sqrt{2\frac{1+r}{1-r}} - \frac{1-r}{8}\sqrt{2\frac{1+r}{1-r}} + \frac{3+r}{8}\sqrt{2\frac{1-r}{3+r}} - \frac{3+r}{8}\sqrt{2\frac{1-r}{3+r}}$$

$$= 0$$

$$\sigma_R^2 = \sigma_{\tilde{R}}^2 = \frac{1-r}{8}2\frac{1+r}{1-r} + \frac{1-r}{8}2\frac{1+r}{1-r} + \frac{3+r}{8}2\frac{1-r}{3+r} + \frac{3+r}{8}2\frac{1-r}{3+r} = 1$$

$$\mathbb{E}_{\mu_{\pi_{\text{ref}}}}\left[R(s,a)\tilde{R}(s,a)\right]$$

$$= \frac{1-r}{8}2\frac{1+r}{1-r} + \frac{1-r}{8}2\frac{1+r}{1-r} - \frac{3+r}{8}2\frac{1-r}{3+r} - \frac{3+r}{8}2\frac{1-r}{3+r} = r.$$

Thus, $\tilde{R}$ is an $r$-correlated proxy.

Since the rewards for both actions are identical at $s_2$ and $s_3$, a policy can only differ meaningfully from $\pi_{\text{ref}}$ at $s_1$. Define

$$\pi_\Delta(a_1 \mid s_1) = \frac{1}{2} + \Delta \qquad \pi_\Delta(a_1 \mid s_2) = \pi_\Delta(a_1 \mid s_3) = 1.$$

as any such policy where $\Delta \in [-1/2, 1/2]$. Then, we can calculate the lower bound $L(\pi_\Delta)$ as

$$L(\pi_\Delta) = \frac{1}{r}\left(J(\pi_\Delta, \tilde{R}) - \sqrt{(1-r^2)\chi^2(\mu_{\pi_\Delta}\|\mu_{\pi_{\text{ref}}})}\right)$$

$$= \frac{1}{r}\left(2\Delta\left(\frac{1-r}{8}\right)\sqrt{2\frac{1+r}{1-r}} - \sqrt{(1-r^2)\frac{1}{2}(1-r)\Delta^2}\right)$$

$$= \frac{1}{r}\left(\Delta\sqrt{\frac{(1+r)(1-r)}{2}} - |\Delta|\sqrt{\frac{(1-r)^2(1+r)}{2}}\right)$$

$$= \frac{1}{r}\sqrt{\frac{(1+r)(1-r)}{2}}\left(\Delta - \sqrt{1-r}|\Delta|\right).$$

Since $\sqrt{1-r} < 1$, clearly this is maximized at $\Delta = 1/2$, where

$$L(\pi_\Delta) = \frac{1}{r}\sqrt{\frac{(1+r)(1-r)}{2}}\left(\frac{1}{2} - \sqrt{1-r}\frac{1}{2}\right) > 0.$$

Furthermore, $\pi_{1/2}$ is an optimal policy with respect to the true reward function, since $R(s_1, a_1) > R(s_1, a_2)$ and $\pi_{1/2}(a_1 \mid s_1) = 1$. Thus, letting $\pi^* = \pi_{1/2}$ completes the proof. $\qquad\square$

## A.2   FAILURE OF ACTION DISTRIBUTION REGULARIZATION

As discussed in the main text, the OM regularization method we propose differs from the AD-based regularization found in previous work on RLHF. The following theorem shows that almost *any* form of policy optimization with action distribution regularization cannot guarantee an improvement in true reward over the reference policy.

**Theorem A.5.** *Fix $r \in (0, 1)$. Consider a policy optimization objective regularized by any $f$-divergence between the action distributions of the learned policy and the reference policy:*

$$\text{maximize} \qquad L'(\pi) = J(\pi, \tilde{R}) - J(\pi_{ref}, \tilde{R}) - g\left((1-\gamma)\mathbb{E}_\pi\left[\sum_{t=0}^{\infty}\gamma^t D_f\left(\pi(\cdot \mid s_t) \| \pi_{ref}(\cdot \mid s_t)\right)\right]\right),$$

*where $f : [0, \infty) \to \mathbb{R}$ is a convex, continuous function with $f(1) = 0$, $g : [0, \infty) \to [0, \infty)$ is a*

*strictly increasing function with $g(0) = 0$, and $D_f$ is the $f$-divergence:*

$$D_f(P \parallel Q) = \mathbb{E}_{x \sim Q}\left[f\left(\frac{P(x)}{Q(x)}\right)\right].$$

*Then there is an MDP, reward functions $R, \tilde{R}$, and reference policy $\pi_{ref}$ such that $\tilde{R}$ is an $r$-correlated proxy for $R$, but there is a policy $\tilde{\pi}$ such that*

$$L'(\tilde{\pi}) > 0 \qquad but \qquad J(\tilde{\pi}, R) < J(\pi_{ref}, R).$$

Theorem A.5 concerns a general form of policy optimization with action distribution regularization; it considers *any* $f$-divergence between action distributions and *any* way of scaling the $f$-divergence using the function $g$. For instnce, $g$ could incorporate linear scaling of the divergence as in the KL regularization used in previous work, square-root scaling as we use with $\chi^2$ divergence in Theorem 5.1, or any other scaling. The theorem shows that no matter how the regularization is formulated, it cannot guarantee improvement in true reward over the reference policy; there is a policy that increases the regularized objective but decreases the true reward.

*Proof.* Define the inverse $g^{-1} : [0, \infty) \to [0, \infty]$ as

$$g^{-1}(x) = \sup \{y \in [0, \infty) \mid g(y) \leq x\}.$$

Since $f$ is continuous and $f(1) = 0$, there must be a radius $\rho > 0$ such that

$$|u - 1| \leq \rho \qquad \Rightarrow \qquad f(u) < \frac{2g^{-1}\left(\frac{1-r}{8}\right)}{1 - r}.$$

We construct an MDP with discount factor

$$\gamma = \begin{cases} \max\left\{1 - \frac{2g^{-1}(\frac{1-r}{8})}{(1-r)f(2)}, \frac{1}{1+\rho}, \frac{1}{2}\right\} & f(2) > 0 \\ \max\left\{\frac{1}{1+\rho}, \frac{1}{2}\right\} & \text{otherwise.} \end{cases}$$

There are four states and two actions. The initial state distribution and transition probabilities are

$$\mu_0(s_1) = \frac{1+r}{4} \qquad\qquad p(s_1 \mid s_1, a_1) = 1 \qquad\qquad p(s_1 \mid s_1, a_2) = 1$$

$$\mu_0(s_2) = \frac{1+r}{4} \qquad\qquad p(s_2 \mid s_2, a_1) = 1 \qquad\qquad p(s_2 \mid s_2, a_2) = 1$$

$$\mu_0(s_3) = \frac{(1-r)(1+\gamma)}{4} \qquad p(s_3 \mid s_3, a_1) = 1 \qquad\qquad p(s_4 \mid s_3, a_2) = 1$$

$$\mu_0(s_4) = \frac{(1-r)(1-\gamma)}{4} \qquad p(s_4 \mid s_4, a_1) = 1 \qquad\qquad p(s_4 \mid s_4, a_2) = 1.$$

The reward functions only depend on the state and are given by

$$\begin{aligned} R(s_1, \cdot) &= 1 & \tilde{R}(s_1, \cdot) &= 1 \\ R(s_2, \cdot) &= -1 & \tilde{R}(s_2, \cdot) &= -1 \\ R(s_3, \cdot) &= 1 & \tilde{R}(s_3, \cdot) &= -1 \\ R(s_4, \cdot) &= -1 & \tilde{R}(s_4, \cdot) &= 1. \end{aligned}$$

In graphical form, the MDP is as follows:

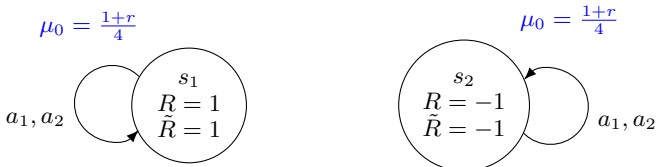

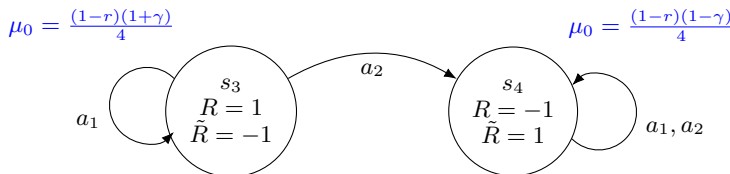

We consider the reference policy defined by

$$\pi_{\text{ref}}(a_1 \mid s_1) = 1 \qquad \pi_{\text{ref}}(a_2 \mid s_1) = 0$$
$$\pi_{\text{ref}}(a_1 \mid s_2) = 1 \qquad \pi_{\text{ref}}(a_2 \mid s_2) = 0$$
$$\pi_{\text{ref}}(a_1 \mid s_3) = \gamma \qquad \pi_{\text{ref}}(a_2 \mid s_3) = 1 - \gamma$$
$$\pi_{\text{ref}}(a_1 \mid s_4) = 1 \qquad \pi_{\text{ref}}(a_2 \mid s_4) = 0.$$

To compute the occupancy measure of the reference policy, we first can see that clearly,

$$\mu_{\pi_{\text{ref}}}(s_1, a_1) = \mu_{\pi_{\text{ref}}}(s_2, a_1) = \frac{1 + r}{4}.$$

For $s_3$, the agent stays in the state until it takes action $a_2$, so we have

$$\mu_{\pi_{\text{ref}}}(s_3) = (1 - \gamma)\mu_0(s_3) \left[ 1 + \gamma^2 + \gamma^4 + \cdots \right]$$
$$= (1 - \gamma)\frac{(1 - r)(1 + \gamma)}{4} \frac{1}{1 - \gamma^2}$$
$$= \frac{1 - r}{4}.$$

This also implies that $\mu_{\pi_{\text{ref}}}(s_4) = \frac{1-r}{4}$ since the occupancy measure sums to one. Thus, we can compute

$$J(\pi_{\text{ref}}, R) = 0 \qquad\qquad J(\pi_{\text{ref}}, \tilde{R}) = 0$$
$$\sigma_R = 1 \qquad\qquad \sigma_{\tilde{R}} = 1$$
$$\mathbb{E}_{\mu_{\pi_{\text{ref}}}}[R(s, a)\tilde{R}(s, a)] = r.$$

This confirms that $\tilde{R}$ is an $r$-correlated proxy for $R$.

Now, we define $\widetilde{\pi}$ as

$$\pi_{\text{ref}}(a_1 \mid s_1) = 1 \qquad \pi_{\text{ref}}(a_2 \mid s_1) = 0$$
$$\pi_{\text{ref}}(a_1 \mid s_2) = 1 \qquad \pi_{\text{ref}}(a_2 \mid s_2) = 0$$
$$\pi_{\text{ref}}(a_1 \mid s_3) = 2\gamma - 1 \qquad \pi_{\text{ref}}(a_2 \mid s_3) = 2(1 - \gamma)$$
$$\pi_{\text{ref}}(a_1 \mid s_4) = 1 \qquad \pi_{\text{ref}}(a_2 \mid s_4) = 0.$$

Note that since $\gamma \geq 1/2$ by definition, the policy is well-defined. As for $\pi_{\text{ref}}$, the occupancy measure

of $\widetilde{\pi}$ at $s_1$ and $s_2$ is $\frac{1+r}{4}$. For $s_3$, we have

$$\mu_{\widetilde{\pi}}(s_3) = (1 - \gamma)\mu_0(s_3) \left[ 1 + \gamma(2\gamma - 1) + \gamma^2(2\gamma - 1)^2 + \cdots \right]$$

$$= (1 - \gamma)\frac{(1 - r)(1 + \gamma)}{4} \frac{1}{1 - \gamma(2\gamma - 1)}$$

$$= \frac{(1 - r)(1 + \gamma)}{4(1 + 2\gamma)}.$$

For $s_4$,

$$\mu_{\widetilde{\pi}}(s_4) = 1 - \mu_{\widetilde{\pi}}(s_1) - \mu_{\widetilde{\pi}}(s_2) - \mu_{\widetilde{\pi}}(s_3)$$

$$= \frac{1 - r}{2} - \frac{(1 - r)(1 + \gamma)}{4(1 + 2\gamma)}$$

$$= \frac{(1 - r)(1 + 3\gamma)}{4(1 + 2\gamma)}.$$

Based on this, we can compute the true reward of $\widetilde{\pi}$:

$$J(\widetilde{\pi}, R) = \mathbb{E}_{\mu_{\widetilde{\pi}}} [R(s, a)]$$

$$= \frac{1 + r}{4} - \frac{1 + r}{4} + \frac{(1 - r)(1 + \gamma)}{4(1 + 2\gamma)} - \frac{(1 - r)(1 + 3\gamma)}{4(1 + 2\gamma)}$$

$$= -\frac{\gamma(1 - r)}{2(1 + 2\gamma)}$$

$$\leq 0 = J(\pi_{\text{ref}}, R).$$

This verifies the claim that $J(\widetilde{\pi}, R) < J(\pi_{\text{ref}}, R)$.

To show the second claim, we can compute the regularized objective $L'(\widetilde{\pi})$. Starting with the proxy reward term, we have

$$J(\widetilde{\pi}, \tilde{R}) = \frac{1 + r}{4} - \frac{1 + r}{4} - \frac{(1 - r)(1 + \gamma)}{4(1 + 2\gamma)} + \frac{(1 - r)(1 + 3\gamma)}{4(1 + 2\gamma)}$$

$$= \frac{\gamma(1 - r)}{2(1 + 2\gamma)}$$

$$\geq \frac{1 - r}{8}. \tag{15}$$

Next, we compute the regularization term, which can be written as

$$g \left( (1 - \gamma)\mathbb{E}_{\widetilde{\pi}} \left[ \sum_{t=0}^{\infty} \gamma^t D_f \Big( \widetilde{\pi}(\cdot \mid s_t) \, \| \, \pi_{\text{ref}}(\cdot \mid s_t) \Big) \right] \right)$$

$$= g \left( \mu(s_3) D_f \Big( \widetilde{\pi}(\cdot \mid s_3) \, \| \, \pi_{\text{ref}}(\cdot \mid s_3) \Big) \right)$$

since $\pi_{\text{ref}}$ and $\widetilde{\pi}$ only differ in state $s_3$. We can rewrite the above as

$$= g \left( \frac{(1 - r)(1 + \gamma)}{4(1 + 2\gamma)} \left[ \pi_{\text{ref}}(a_1 \mid s_3)f \left( \frac{\widetilde{\pi}(a_1 \mid s_3)}{\pi_{\text{ref}}(a_1 \mid s_3)} \right) + \pi_{\text{ref}}(a_2 \mid s_3)f \left( \frac{\widetilde{\pi}(a_2 \mid s_3)}{\pi_{\text{ref}}(a_2 \mid s_3)} \right) \right] \right)$$

$$= g \left( \frac{(1 - r)(1 + \gamma)}{4(1 + 2\gamma)} \left[ \gamma f \left( 2 - \frac{1}{\gamma} \right) + (1 - \gamma)f(2) \right] \right). \tag{16}$$

Note that since $\gamma \geq \frac{1}{1+\rho}$, we have

$$\frac{1}{\gamma} \leq 1 + \rho$$

$$2 - \frac{1}{\gamma} \geq 1 - \rho$$

$$f\left(2 - \frac{1}{\gamma}\right) < \frac{2g^{-1}\left(\frac{1-r}{8}\right)}{1-r}.$$

Plugging this back into (16) along with the fact that $(1-\gamma)f(2) \leq \frac{2g^{-1}\left(\frac{1-r}{8}\right)}{(1-r)}$, we get

$$< g\left(\frac{(1-r)(1+\gamma)}{4(1+2\gamma)}\left[\gamma\frac{2g^{-1}\left(\frac{1-r}{8}\right)}{1-r} + \frac{2g^{-1}\left(\frac{1-r}{8}\right)}{1-r}\right]\right)$$

$$\leq g\left(\frac{1-r}{4} \times \frac{4g^{-1}\left(\frac{1-r}{8}\right)}{1-r}\right)$$

$$\leq \frac{1-r}{8}.$$

Combining this with (15), we have

$$L'(\pi) = J(\pi, \tilde{R}) - J(\pi_{\text{ref}}, \tilde{R}) - g\left((1-\gamma)\mathbb{E}_\pi\left[\sum_{t=0}^{\infty}\gamma^t D_f\left(\pi(\cdot \mid s_t) \parallel \pi_{\text{ref}}(\cdot \mid s_t)\right)\right]\right)$$

$$> \frac{1-r}{8} - 0 - \frac{1-r}{8} = 0,$$

which completes the proof. $\qquad\square$

## A.3 ACTION DISTRIBUTION AND OCCUPANCY MEASURE DIVERGENCES IN LLMS

As noted in the main text, in the current paradigm of using RLHF to train LLMs, we can show that action distribution divergence between two policies is equivalent to occupancy measure divergence. In particular, RLHF for LLMs is usually modeled as a *contextual bandit*.

In our setting, a contextual bandit can be defined as an MDP with $\gamma = 0$; then, the return of the policy $\pi$ under a reward function $R$ is given by

$$J(\pi, R) = \mathbb{E}_{s\sim\mu_0(\cdot), a\sim\pi(\cdot|s)}[R(s,a)].$$

That is, a single state is sampled from the initial state distribution $\mu_0$, and then a single action is sampled from the policy $\pi$ conditioned on that state. RLHF for LLMs follows this setting as a prompt is sampled from a dataset, the LLM generates a response, and the reward is calculated based on the prompt and response.

In this setting, it is simple to show that the action distribution and occupancy measure divergences are equivalent.

**Lemma A.6.** *Let $D_f(P \parallel Q) = \mathbb{E}_{x\sim Q}\left[f\left(P(x)/Q(x)\right)\right]$ be the $f$-divergence between two distributions $P$ and $Q$. Then, for any two policies $\pi, \pi'$ in a contextual bandit, we have*

$$D_f(\mu_\pi \parallel \mu_{\pi'}) = \mathbb{E}_{s\sim\mu_0(\cdot)}\left[D_f\left(\pi(\cdot \mid s) \parallel \pi'(\cdot \mid s)\right)\right].$$

Lemma A.6 applies to any $f$-divergence, including the KL and $\chi^2$ divergences we study in this paper.

*Proof.* We have

$$
\begin{aligned}
D_f(\mu_\pi \parallel \mu_{\pi'}) &= \mathbb{E}_{s\sim\mu_0(\cdot),a\sim\pi'(\cdot|s)}\left[f\left(\frac{\mu_\pi(s,a)}{\mu_{\pi'}(a\mid s)}\right)\right] \\
&= \mathbb{E}_{s\sim\mu_0(\cdot)}\left[\mathbb{E}_{a\sim\pi'(\cdot|s)}\left[f\left(\frac{\mu_0(s)\pi(a\mid s)}{\mu_0(s)\pi'(a\mid s)}\right)\right]\right] \\
&= \mathbb{E}_{s\sim\mu_0(\cdot)}\left[D_f(\pi \parallel \pi')\right],
\end{aligned}
$$

which completes the proof. $\qquad\square$

### A.3.1 AUTOREGRESSIVE ENVIRONMENTS

While most RLHF implementations use the contextual bandit formulation above for the purposes of KL regularization, one can also model training an LLM as a sequential problem where each token generated is a separate action. This formulation is no longer a contextual bandit, but we can show that the action distribution and occupancy measure KL divergences are still equivalent!

**Lemma A.7.** *Suppose that an environment satisfies the following conditions:*

- *It is deterministic: $\mu_0(s_0) = 1$ for exactly one state $s_0$, and for all $s_t, a_t \in \mathcal{S} \times \mathcal{A}$, $p(s_{t+1} \mid s_t, a_t) = 1$ for exactly one state $s_{t+1}$.*

- *Exactly one sequence of actions leads to each state: if following $a_0, \ldots, a_{t-1}$ leads to $s$, then no other sequence of actions (of any length) can also lead to $s$.*

*Then, for any policies $\pi, \pi'$, the action distribution and occupancy measure KL divergences between them are equal (removing the $(1-\gamma)$ prefactor on the action distribution divergence):*

$$
D_{KL}(\mu_\pi \parallel \mu_{\pi'}) = \mathbb{E}_\pi\left[\sum_{t=0}^\infty \gamma^t D_{KL}(\pi(\cdot \mid s_t) \parallel \pi'(\cdot \mid s_t))\right].
$$

Lemma A.7 applies to LLMs since one can treat the "state" of the environment after $t$ timesteps as all the tokens generated so far $w_0 w_1 \ldots w_{t-1}$, and the actions as the next token $w_t$, which is then appended to the state:

$$a_t \sim \pi(a_t \mid s_t) = \pi(w_t \mid w_0 w_1 \ldots w_{t-1}).$$
$$p(s_{t+1} \mid s_t, a_t) = \mathbf{1}\{s_{t+1} = w_0 w_1 \ldots w_{t-1} w_t\} \qquad \text{where} \quad s_t = w_0 w_1 \ldots w_{t-1} \quad \text{and} \quad a_t = w_t.$$

Thus, regardless of the formalism used to train an LLM via RLHF, the action distribution and occupancy measure KL are equivalent. However, the conditions of Lemma A.7 are unlikely to be met by many other MDPs. Many MDPs are stochastic, violating the first assumption. Even among deterministic MDPs, it is very uncommon that only a single action sequence can lead to each state.

*Proof.* Given the assumptions about the environment, we can rewrite the log-occupancy measure of a state-action pair in terms of the sum of log action probabilties over the unique sequence of actions leading to that state. Suppose $a_0, \ldots, a_{t-1}$ is the unique action sequence leading to $s$ and that this action sequence visits states $s_0, \ldots, s_{t-1}, s$. Then

$$
\begin{aligned}
\log\mu_\pi(s,a) &= \log\left((1-\gamma)\mathbb{E}_\pi\left[\sum_{t=0}^\infty \gamma^t \mathbf{1}\{s_t = s \wedge a_t = a\}\right]\right) \\
&= \log\left((1-\gamma)\gamma^t \mathbb{P}_\pi(s_t = s \wedge a_t = a)\right) \\
&= \log\left((1-\gamma)\gamma^t \prod_{i=0}^t \pi(a_i \mid s_i)\right) \\
&= \log(1-\gamma) + t\log\gamma + \sum_{i=0}^t \log\pi(a_i \mid s_i).
\end{aligned}
$$

Using this, we can rewrite the occupancy measure KL divergence as

$$
\begin{aligned}
D_{\mathrm{KL}}(\mu_\pi \parallel \mu_{\pi'}) &= \sum_{(s,a)\in\mathcal{S}\times\mathcal{A}} \mu_\pi(s,a) \log\left(\frac{\mu_\pi(s,a)}{\mu_{\pi'}(s,a)}\right) \\
&= (1-\gamma)\sum_{t=0}^{\infty}\gamma^t \sum_{a_0,\ldots,a_t\in\mathcal{A}^{t+1}} \mathbb{P}_\pi(a_0\wedge\cdots\wedge a_t)\sum_{i=0}^{t}\Big(\log\pi(a_i\mid s_i)-\log\pi'(a_i\mid s_i)\Big) \\
&= (1-\gamma)\sum_{t=0}^{\infty}\gamma^t \sum_{a_0,\ldots,a_t\in\mathcal{A}^{t+1}} \left(\prod_{j=0}^{t}\pi(a_i\mid s_i)\right)\sum_{i=0}^{t}\Big(\log\pi(a_i\mid s_i)-\log\pi'(a_i\mid s_i)\Big),
\end{aligned}
\tag{17}
$$

where $s_i$ is the state reached by taking $a_0,\ldots,a_{i-1}$.

We will now show inductively that

$$
\sum_{a_0,\ldots,a_t\in\mathcal{A}^{t+1}} \left(\prod_{j=0}^{t}\pi(a_j\mid s_j)\right)\sum_{i=0}^{t}\Big(\log\pi(a_i\mid s_i)-\log\pi'(a_i\mid s_i)\Big)
$$
$$
= \sum_{i=0}^{t}\sum_{s_i\in\mathcal{S}} \mathbb{P}_\pi(s_i)D_{\mathrm{KL}}(\pi(\cdot\mid s_i)\parallel\pi'(\cdot\mid s_i)).
\tag{18}
$$

Consider first if $t=0$. Then

$$
\sum_{a_0\in\mathcal{A}}\pi(a_0\mid s_0)\Big(\log\pi(a_0\mid s_0)-\log\pi'(a_0\mid s_0)\Big)
$$
$$
= D_{\mathrm{KL}}(\pi(\cdot\mid s_0)\parallel\pi'(\cdot\mid s_0))
$$
$$
= \mathbb{P}_\pi(s_0)D_{\mathrm{KL}}(\pi(\cdot\mid s_0)\parallel\pi'(\cdot\mid s_0)).
$$

Now suppose (18) holds for $t - 1$. Then for $t$ we have

$$\sum_{a_0,\ldots,a_t \in \mathcal{A}^{t+1}} \left( \prod_{j=0}^{t} \pi(a_j \mid s_j) \right) \sum_{i=0}^{t} \left( \log \pi(a_i \mid s_i) - \log \pi'(a_i \mid s_i) \right)$$

$$= \sum_{a_0,\ldots,a_{t-1} \in \mathcal{A}^{t}} \left( \prod_{j=0}^{t-1} \pi(a_j \mid s_j) \right) \sum_{a_t \in \mathcal{A}} \pi(a_t \mid s_t) \left[ \log \pi(a_t \mid s_t) - \log \pi'(a_t \mid s_t) \right.$$
$$\left. + \sum_{i=0}^{t-1} \left( \log \pi(a_i \mid s_i) - \log \pi'(a_i \mid s_i) \right) \right]$$

$$= \sum_{a_0,\ldots,a_{t-1} \in \mathcal{A}^{t}} \left( \prod_{j=0}^{t-1} \pi(a_j \mid s_j) \right) \left[ D_{\mathrm{KL}}(\pi(\cdot \mid s_t) \parallel \pi'(\cdot \mid s_t)) \right.$$
$$\left. + \sum_{a_t \in \mathcal{A}} \pi(a_t \mid s_t) \sum_{i=0}^{t-1} \left( \log \pi(a_i \mid s_i) - \log \pi'(a_i \mid s_i) \right) \right]$$

$$= \sum_{a_0,\ldots,a_{t-1} \in \mathcal{A}^{t}} \left( \prod_{j=0}^{t-1} \pi(a_j \mid s_j) \right) \left[ D_{\mathrm{KL}}(\pi(\cdot \mid s_t) \parallel \pi'(\cdot \mid s_t)) \right.$$
$$\left. + \sum_{i=0}^{t-1} \left( \log \pi(a_i \mid s_i) - \log \pi'(a_i \mid s_i) \right) \right]$$

$$= \sum_{s_t \in \mathcal{S}} \mathbb{P}_\pi(s_t) D_{\mathrm{KL}}(\pi(\cdot \mid s_t) \parallel \pi'(\cdot \mid s_t))$$
$$+ \sum_{a_0,\ldots,a_{t-1} \in \mathcal{A}^{t}} \left( \prod_{j=0}^{t-1} \pi(a_j \mid s_j) \right) \sum_{i=0}^{t-1} \left( \log \pi(a_i \mid s_i) - \log \pi'(a_i \mid s_i) \right)$$

$$\overset{(i)}{=} \sum_{s_t \in \mathcal{S}} \mathbb{P}_\pi(s_t) D_{\mathrm{KL}}(\pi(\cdot \mid s_t) \parallel \pi'(\cdot \mid s_t)) + \sum_{i=0}^{t-1} \sum_{s_i \in \mathcal{S}} \mathbb{P}_\pi(s_i) D_{\mathrm{KL}}(\pi(\cdot \mid s_i) \parallel \pi'(\cdot \mid s_i))$$

$$= \sum_{i=0}^{t} \sum_{s_i \in \mathcal{S}} \mathbb{P}_\pi(s_i) D_{\mathrm{KL}}(\pi(\cdot \mid s_i) \parallel \pi'(\cdot \mid s_i)),$$

where (i) is from the inductive hypothesis.

Now, plugging (18) into (17) gives

$$D_{\text{KL}}(\mu_\pi \parallel \mu_{\pi'})$$

$$= (1-\gamma) \sum_{t=0}^{\infty} \gamma^t \sum_{i=0}^{t} \sum_{s_i \in \mathcal{S}} \mathbb{P}_\pi(s_i) D_{\text{KL}}(\pi(\cdot \mid s_i) \parallel \pi'(\cdot \mid s_i))$$

$$= (1-\gamma) \sum_{i=0}^{\infty} \sum_{t=i}^{\infty} \gamma^t \sum_{s_i \in \mathcal{S}} \mathbb{P}_\pi(s_i) D_{\text{KL}}(\pi(\cdot \mid s_i) \parallel \pi'(\cdot \mid s_i))$$

$$= (1-\gamma) \sum_{i=0}^{\infty} \sum_{s_i \in \mathcal{S}} \mathbb{P}_\pi(s_i) D_{\text{KL}}(\pi(\cdot \mid s_i) \parallel \pi'(\cdot \mid s_i)) \sum_{t=i}^{\infty} \gamma^t$$

$$= (1-\gamma) \mathbb{E}_\pi \left[ \sum_{i=0}^{\infty} D_{\text{KL}}(\pi(\cdot \mid s_i) \parallel \pi'(\cdot \mid s_i)) \sum_{t=i}^{\infty} \gamma^t \right]$$

$$= (1-\gamma) \mathbb{E}_\pi \left[ \frac{\gamma^i}{1-\gamma} \sum_{i=0}^{\infty} D_{\text{KL}}(\pi(\cdot \mid s_i) \parallel \pi'(\cdot \mid s_i)) \right]$$

$$= \mathbb{E}_\pi \left[ \gamma^i \sum_{i=0}^{\infty} D_{\text{KL}}(\pi(\cdot \mid s_i) \parallel \pi'(\cdot \mid s_i)) \right],$$

which is the desired result. □

### A.4 LEARNED REWARD FUNCTIONS ARE $r$-CORRELATED

Proxy reward functions are often *learned* from data like ratings or preference comparisons, including in the case of RLHF. Here, we show that a learned reward function with low mean-squared error—a common objective in supervised learning—is $r$-correlated with the true reward function.

**Lemma A.8.** *Let $R$ be the true reward function and $\tilde{R}$ be a learned reward function. Suppose that $\mathbb{E}_{\mu_{\pi_{\text{ref}}}} \left[ (R(s,a) - \tilde{R}(s,a))^2 \right] \leq \epsilon \sigma_R^2$. Then, the learned reward function is an $r$-correlated proxy with $r \geq 1 - \epsilon$.*

The assumption in Lemma A.8 is that the mean-squared error over the occupancy measure of the reference policy is small. This can be achieved, for example, by learning $\tilde{R}$ via least-squares regression over a training dataset of state-action pairs sampled from the reference policy. Many results in learning theory show that this results in a small mean-squared error over the distribution the training data was sampled from, i.e., exactly the assumption in Lemma A.8 (Koltchinskii, 2006).

*Proof.* Throughout the proof, all expectations, variances, and covariances are with respect to $\mu_{\pi_{\text{ref}}}$. We can rewrite the assumption using the bias-variance decomposition as

$$\mathbb{E}\left[ (R(s,a) - \tilde{R}(s,a))^2 \right]$$

$$= \text{Var}\left[ R(s,a) - \tilde{R}(s,a) \right] + \left( \mathbb{E}\left[ R(s,a) \right] - \mathbb{E}\left[ \tilde{R}(s,a) \right] \right)^2$$

$$= \text{Var}\left[ R(s,a) \right] + \text{Var}\left[ \tilde{R}(s,a) \right] - 2\,\text{Cov}\left[ R(s,a), \tilde{R}(s,a) \right] + \left( \mathbb{E}\left[ R(s,a) \right] - \mathbb{E}\left[ \tilde{R}(s,a) \right] \right)^2$$

$$= \sigma_R^2 + \sigma_{\tilde{R}}^2 - 2\,\text{Cov}\left[ R(s,a), \tilde{R}(s,a) \right] + \left( \mathbb{E}\left[ R(s,a) \right] - \mathbb{E}\left[ \tilde{R}(s,a) \right] \right)^2$$

$$\leq \epsilon \sigma_R^2.$$

Note that $\left( \mathbb{E}\left[ R(s,a) \right] - \mathbb{E}\left[ \tilde{R}(s,a) \right] \right)^2 > 0$, so we can rewrite the inequality as

$$2\text{Cov}\left[ R(s,a), \tilde{R}(s,a) \right] \geq (1-\epsilon)\sigma_R^2 + \sigma_{\tilde{R}}^2 \geq (1-\epsilon)\left( \sigma_R^2 + \sigma_{\tilde{R}}^2 \right).$$

Dividing both sides by $2\sigma_R\sigma_{\tilde{R}}$ gives

$$\frac{\text{Cov}\left[R(s,a),\tilde{R}(s,a)\right]}{\sigma_R\sigma_{\tilde{R}}} \geq \frac{1-\epsilon}{2}\frac{\sigma_R^2+\sigma_{\tilde{R}}^2}{\sigma_R\sigma_{\tilde{R}}}.$$

By the AM-GM inequality, $\frac{\sigma_R^2+\sigma_{\tilde{R}}^2}{2} \geq \sigma_R\sigma_{\tilde{R}}$, so

$$\frac{\text{Cov}\left[R(s,a),\tilde{R}(s,a)\right]}{\sigma_R\sigma_{\tilde{R}}} \geq 1-\epsilon,$$

which is the desired result. $\qquad\square$

### A.5 State-Only Occupancy Measures

We can also consider state-only occupancy measures, defined as

$$\mu_\pi(s) = (1-\gamma)\mathbb{E}_\pi\left[\sum_{t=0}^\infty \gamma^t \mathbb{1}\{s_t = s\}\right].$$

In many environments, the reward functions only depend on the state, i.e., $R(s,a) = R(s)$ and $\tilde{R}(s,a) = \tilde{R}(s)$. In this case, Theorem 5.1 holds for state-only occupancy measures as well. The proof is identical to the proof of Theorem 5.1, but replacing expectations and sums over state-action pairs with expectations over states.

## B Derivation of Occupancy-Regularized Policy Optimization

In this appendix section, we show how to derive the approximations used for Occupancy-Regularized Policy Optimization (ORPO). As a reminder, we would like to optimize

$$J(\pi_\theta, \tilde{R}) - \lambda\sqrt{\chi^2\left(\mu_{\pi_\theta}\|\mu_{\pi_{\text{ref}}}\right)}.$$

We can rewrite its gradient as

$$\nabla_\theta\left(J(\pi_\theta, \tilde{R}) - \lambda\sqrt{\chi^2\left(\mu_{\pi_\theta}\|\mu_{\pi_{\text{ref}}}\right)}\right)$$

$$= \nabla_\theta J(\pi_\theta, \tilde{R}) - \frac{\lambda\nabla_\theta\chi^2\left(\mu_{\pi_\theta}\|\mu_{\pi_{\text{ref}}}\right)}{2\sqrt{\chi^2\left(\mu_{\pi_\theta}\|\mu_{\pi_{\text{ref}}}\right)}}$$

$$= \nabla_\theta\left(\sum_{s,a}\mu_{\pi_\theta}(s,a)\tilde{R}(s,a)\right) - \frac{\lambda}{2\sqrt{\chi^2\left(\mu_{\pi_\theta}\|\mu_{\pi_{\text{ref}}}\right)}}\nabla_\theta\left(\sum_{s,a}\frac{\mu_{\pi_\theta}(s,a)^2}{\mu_{\pi_{\text{ref}}}(s,a)} - 1\right)$$

$$= \sum_{s,a}\left[\nabla_\theta\mu_{\pi_\theta}(s,a)\tilde{R}(s,a) - \frac{\lambda}{2\sqrt{\chi^2\left(\mu_{\pi_\theta}\|\mu_{\pi_{\text{ref}}}\right)}}\nabla_\theta\left(\frac{\mu_{\pi_\theta}(s,a)^2}{\mu_{\pi_{\text{ref}}}(s,a)} - 1\right)\right]$$

$$= \sum_{s,a}\left[\nabla_\theta\mu_{\pi_\theta}(s,a)\tilde{R}(s,a) - \frac{\lambda}{2\sqrt{\chi^2\left(\mu_{\pi_\theta}\|\mu_{\pi_{\text{ref}}}\right)}}\frac{2\mu_{\pi_\theta}(s,a)}{\mu_{\pi_{\text{ref}}}(s,a)}\nabla_\theta\mu_{\pi_\theta}(s,a)\right]$$

$$= \sum_{s,a}\left(\nabla_\theta\mu_{\pi_\theta}(s,a)\right)\left(\tilde{R}(s,a) - \frac{\lambda}{\sqrt{\chi^2(\mu_{\pi_\theta}\|\mu_{\pi_{\text{ref}}})}}\frac{\mu_{\pi_\theta}(s,a)}{\mu_{\pi_{\text{ref}}}(s,a)}\right).$$

As described in the main text, policy gradient algorithms can approximate this type of gradient by using an augmented reward function

$$R'(s,a) = \tilde{R}(s,a) - \frac{\lambda}{\sqrt{\chi^2(\mu_{\pi_\theta}\|\mu_{\pi_{\text{ref}}})}}\frac{\mu_{\pi_\theta}(s,a)}{\mu_{\pi_{\text{ref}}}(s,a)}. \tag{19}$$

However, two terms in (19) cannot be computed directly: the current $\chi^2$ divergence between occupancy measures, and the ratio of the occupancy measures $\mu_{\pi_\theta}(s,a)/\mu_{\pi_{\text{ref}}}(s,a)$. We first show how to approximate the latter using a discriminator network $\hat{d}_\phi(s,a)$, trained to optimize

$$\phi = \arg\min_\phi \; \mathbb{E}_{\mu_{\pi_\theta}}\left[ \log(1 + e^{-\hat{d}_\phi(s,a)}) \right] + \mathbb{E}_{\mu_{\pi_{\text{ref}}}}\left[ \log(1 + e^{\hat{d}_\phi(s,a)}) \right]. \tag{20}$$

It is well-known that exactly optimizing the loss function in (20) gives

$$\hat{d}_\phi(s,a) = \log \frac{\mu_{\pi_\theta}(s,a)}{\mu_{\pi_{\text{ref}}}(s,a)}. \tag{21}$$

Furthermore, the OM $\chi^2$ divergence is given by

$$\chi^2\left(\mu_{\pi_\theta} \| \mu_{\pi_{\text{ref}}}\right) = \mathbb{E}_{\mu_{\pi_\theta}}\left[ \frac{\mu_{\pi_\theta}(s,a)}{\mu_{\pi_{\text{ref}}}(s,a)} - 1 \right] = \mathbb{E}_{\mu_{\pi_\theta}}\left[ e^{\hat{d}_\phi(s,a)} - 1 \right] =: \widehat{\chi^2}. \tag{22}$$

Combining (21) and (22) shows that the augmented reward in (19) can be rewritten as

$$R'(s,a) = \tilde{R}(s,a) - \frac{\lambda}{\sqrt{\widehat{\chi^2}}} e^{\hat{d}_\phi(s,a)}.$$

Putting all the steps together, the following algorithm formalizes ORPO:

---

**Algorithm 1** Occupancy-Regularized Policy Optimization (ORPO).

---

1: **for** iteration $i = 1, \ldots, I$ **do**
2:     Collect a set of $n$ trajectories $\mathcal{D}_\pi$ from $\pi_\theta$.
3:     Collect a set of $n$ trajectories $\mathcal{D}_{\pi_{\text{ref}}}$ from $\pi_{\text{ref}}$.
4:     Optimize $\phi$ via SGD to minimize

$$\mathbb{E}_{\mathcal{D}_\pi}\left[ \log(1 + e^{-\hat{d}_\phi(s,a)}) \right] + \mathbb{E}_{\mathcal{D}_{\pi_{\text{ref}}}}\left[ \log(1 + e^{\hat{d}_\phi(s,a)}) \right]$$

5:     Calculate $\widehat{\chi^2} = \mathbb{E}_{\mathcal{D}_\pi}\left[ e^{\hat{d}_\phi(s,a)} - 1 \right]$.
6:     Transform $\mathcal{D}_\pi$ to $\mathcal{D}'_\pi$ by replacing the rewards with $R'(s,a) = \tilde{R}(s,a) - \frac{\lambda}{\sqrt{\widehat{\chi^2}}}\left( e^{\hat{d}_\phi(s,a)} - 1 \right)$.
7:     Optimize $\theta$ via SGD to minimize the proximal policy optimization (PPO) loss $L_{\text{PPO}}\left(\mathcal{D}'_\pi\right)$.
8: **end for**

---

A similar approach can also be used to optimize the proxy reward regularized by KL divergence

$$J(\pi_\theta, \tilde{R}) - \lambda \, D_{\text{KL}}\left(\mu_{\pi_\theta} \| \mu_{\pi_{\text{ref}}}\right),$$

by changing the augmented reward in Line 6 of Algorithm 1 to

$$R'(s,a) = \tilde{R}(s,a) - \lambda \, \hat{d}_\phi(s,a).$$

**How accurate is the discriminator-based approximation?.** To determine whether using the discriminator results in an accurate approximation of $\chi^2$ and $KL$ OM divergences, we plot the output of the discriminator in the glucose environment versus the theoretically correct value in Figure 6. The results suggest that the discriminator accurately approximates the log ratio of the occupancy measures, which in turn allows for accurate approximations of the OM divergences.

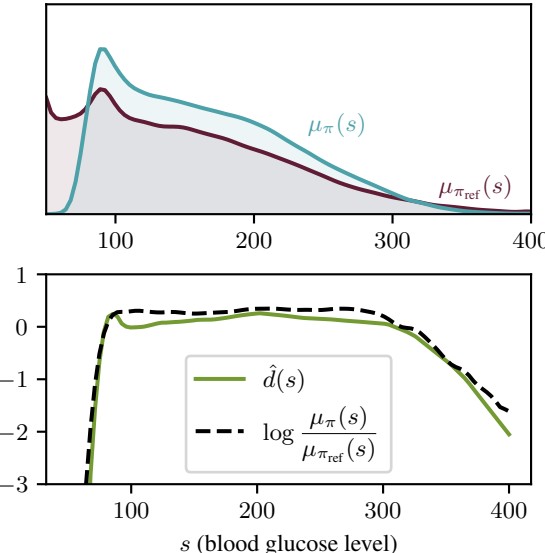

Figure 6: The top plot shows the state occupancy measures of the reference policy $\pi_{\text{ref}}$ and a policy $\pi$ optimized with ORPO in the glucose environment. The bottom plot shows the exact log ratio of the occupancy measures $\log \mu_\pi(s)/\mu_{\pi_{\text{ref}}}(s)$ versus the discriminator output $\hat{d}_\phi(s)$, which attempts to approximate it. We find the discriminator output to be a good approximation of the log ratio.

## C   ENVIRONMENT DETAILS

Here, we discuss the details of the five reward-hacking environments we study.

### C.1   TRAFFIC CONTROL

The traffic control environment, based on the Flow simulator (Wu et al., 2022), simulates a group of human-controlled and RL-controlled vehicles on an on-ramp attempting to merge into traffic on a highway. The true reward prioritizes a small mean commute time, while the proxy reward is the average velocity of all cars. When reward hacking, the RL controlled vehicle on the on-ramp stops indefinitely and lets cars continue forward at high speeds on the highway, which maximizes the proxy reward but increases the commute times of cars on the on-ramp infinitely. As the reference policy for the traffic environment we used the Intelligent Driver Model (IDM), a standard approximation of human driving behavior (Treiber et al., 2000). In practice, reference policies are often learned via imitation learning, so to simulate this we generate data from the IDM controller and train a behavioral cloning (BC) policy using the generated data.

Here, the green cars are controlled by the human driver model IDM controller, and the blue cars are controlled by RL:

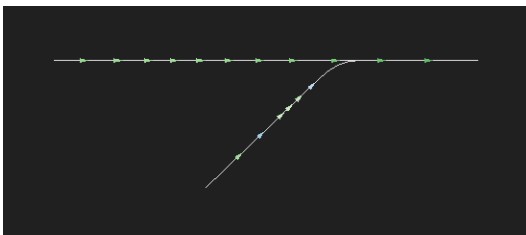

This particular frame showcases reward hacking behavior. The blue RL-controlled vehicle has stopped completely on the on-ramp, blocking cars behind it. This increases the average velocity of all vehicles in the simulation, as the cars on the straightway are able to continue speeding along the road without having to wait for merging cars. However, the true reward (negative average commute

time) decreases endlessly as the cars on the on-ramp wait.

## C.2 GLUCOSE MONITORING

The SimGlucose blood glucose monitoring environment is an extension of the FDA-approved glucose monitoring simulator proposed by Man et al. (2014) for Type 1 Diabetes patients (Fox et al., 2020):

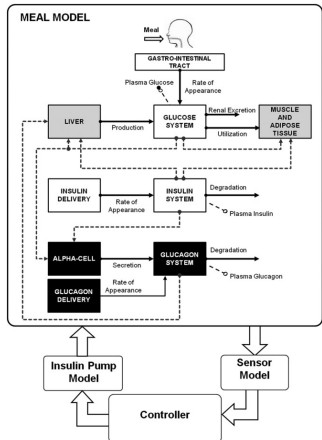

The RL agent (bottom) controls the insulin administered to a simulated patient in order to maintain healthy glucose levels. The true reward is a standard measure of health risk for the patient, but the proxy reward prioritizes the monetary cost of insulin. As the safe baseline policy, we train a BC policy using data generated by a PID controller with parameters tuned by the original designers of the simulator (Steil, 2013).

In Figure 2, we adapt a diagram from Pauley et al. (2022) to represent this environment.

## C.3 PANDEMIC MITIGATION

PandemicSimulator (Kompella et al., 2020) models a population's infection dynamics using a modified SEIR model that they designed to represent the COVID-19 pandemic:

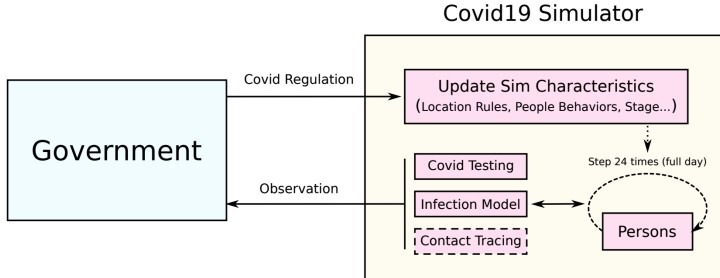

The RL agent chooses the level of lockdown restrictions placed on the population by observing the results of testing. The proxy reward function omits the political cost associated with certain decisions. Our reference policy is trained via BC on a combination of hand-specified and real-world strategies employed by governments during the pandemic, which were also used by Kompella et al. (2020) as baselines.

## C.4 RLHF

We base our RLHF environment on the work of Coste et al. (2024), who study overoptimization of LLM-based reward models. The proxy reward model we use is fine-tuned from Pythia-70M (Biderman et al., 2023) on the AlpacaFarm (Dubois et al., 2023) preference dataset. As a true reward

model, we use the Llama 3 Tulu V2 8B RM from AI2 (Ivison et al., 2024). As a reference policy, we use Coste et al.'s SFT policy, which was fine-tuned from Pythia-1.4B on the AlpacaFarm SFT data. We evaluate policies' true and proxy rewards on responses sampled for a held out set of prompts.

### C.5 TOMATO-WATERING GRIDWORLD

The tomato environment contains a sprinkler state where the agent perceives all tomatoes as being watered and thus receives high proxy reward but no true reward. We train a reference policy using the true reward function, and then add a 10% chance of taking a random action to ensure there is room to improve upon it.

The gray squares in the environment represent walls, and the white squares represent open spaces where the agent can travel:

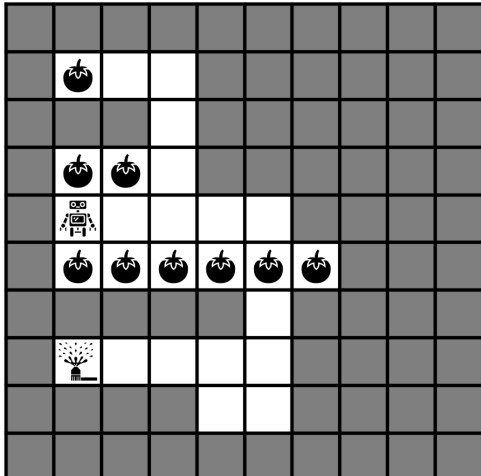

The sprinkler state is down a narrow hallway, and on the other end a tomato is down another narrow hallway.

## D    EXPERIMENT DETAILS

### D.1    NON-LLM EXPERIMENTS

We implement ORPO using RLLib (Liang et al., 2018) and PyTorch (Paszke et al., 2019).

**Network architectures**    For the pandemic, traffic, and tomato-watering environments, we use policy networks based on fully-connected networks with 2 layers of 128 units, 4 layers of 512 units, and 4 units of 512 units, respectively. The policy model for the glucose environment is a basic LSTM network with 3 layers of 64 units each. We made this choice since the observation of the environment contains continuous historical information about the patient's blood glucose levels and previously administered insulin.

The discriminator model for all four non-LLM environments is a fully connected network with 2 layers of 256 units each. We found that the discriminator architecture did not need to be tuned to each environment.

**Policy initialization**    Initializing using an imitation learning policy has been shown to effectively speed up the learning process (Laidlaw et al., 2023; Uchendu et al., 2023) and is used in practice for RLHF (Stiennon et al., 2020), so we initialize our policies using the specified $\pi_{\text{ref}}$ for the more realistic traffic, glucose, and pandemic environments.

**Hyperparameters** Some hyperparameters for the traffic environment were tuned by Pan et al. (2022). We primarily tuned the hyperparameters listed below in order to ensure that the proxy reward would be properly optimized and reward hacking would occur without regularization. This enables to see if the various regularization methods actually succeed at preventing reward hacking.

| Hyperparameter | Tomato | Traffic | Glucose | Pandemic |
|---|---|---|---|---|
| Training iterations | 500 | 250 | 500 | 260 |
| Batch size | 3000 | 40000 | 100000 | 3860 |
| SGD minibatch size | 128 | 16384 | 1024 | 64 |
| SGD epochs per iteration | 8 | 5 | 4 | 5 |
| Optimizer | Adam | Adam | Adam | Adam |
| Learning rate | 1e-3 | 5e-5 | 1e-4 | 0.0003 |
| Gradient clipping | 0.1 | None | 10 | 10 |
| Discount rate ($\gamma$) | 0.99 | 0.99 | 0.99 | 0.99 |
| GAE coefficient ($\lambda$) | 0.98 | 0.97 | 0.98 | 0.95 |
| Entropy coefficient (start) | 0.01 | 0.01 | 0.01 | 0.1 |
| Entropy coefficient (end) | 0.01 | 0.01 | 0.01 | 0.01 |
| Entropy schedule horizon | 0 | 0 | 0 | 500000 |
| KL target | 0.001 | 0.02 | 1e-3 | 0.01 |
| Value function loss clipping | 10 | 10,000 | 100 | 20 |
| Value function loss coefficient | 0.1 | 0.5 | 0.0001 | 0.5 |
| Share value function layers | F | T | T | T |

Table 2: PPO/ORPO hyperparameters.

| Hyperparameter | Tomato | Traffic | Glucose | Pandemic |
|---|---|---|---|---|
| Discriminator reward clipping | 1000 | 10 | 1e10 | 0.1 |
| Regularization coefficient ($\lambda$) | Varied | Varied | Varied | Varied |
| $\sigma_{\tilde{R}}$ | 0.05 | 0.0002 | 0.05 | 0.08 |

Table 3: ORPO-specific hyperparameters.

**ORPO details** We found that a couple of tricks were useful to ensure that ORPO remained stable. First, we clip the discriminator term added to the reward functions to a range $[-\delta, \delta]$, since sometimes it can blow up and cause numerical issues. Second, when estimating $\widehat{\chi^2}$, we use a trimmed mean (trimmed by 1% in each tail) to reduce the effect that outliers have on the estimate. These are both particularly important for $\chi^2$ divergence, where the output of the discriminator is exponentiated for both the reward discriminator term and for estimating $\widehat{\chi^2}$.

## D.2 RLHF Experiments

We train LLMs via the RLHF implementation used by Coste et al. (2024), which is based on OpenAssistant and trlX (Havrilla et al., 2023). To implement $\chi^2$ or KL regularization, we directly add a loss term to the PPO loss:

$$\chi^2 \text{ divergence:} \quad \lambda\left(\frac{\pi_\theta(a \mid s)}{\pi_{\text{ref}}(a \mid s)} + \frac{\pi_{\text{ref}}(a \mid s)}{\pi_\theta(a \mid s)} - 2\right)$$

$$\text{KL divergence:} \quad \lambda\left(\log\frac{\pi_\theta(a \mid s)}{\pi_{\text{ref}}(a \mid s)} + \frac{\pi_{\text{ref}}(a \mid s)}{\pi_\theta(a \mid s)} - 1\right)$$

where $s$ is the prompt and $a$ is the sampled response. Intuitively, both loss terms have a unique minimum when $\pi_\theta(a \mid s) = \pi_{\text{ref}}(a \mid s)$ and in expectation are equivalent to the correct divergence. Schulman (2020) suggests that these are particularly low-variance estimates, and we find that they

work well in practice.

# E  ADDITIONAL EXPERIMENTS AND RESULTS

In this appendix, we the full results from our main experiments as well as ablations of ORPO.

## E.1  WIN RATES FOR RLHF

Besides calculating the true reward for RLHF models with the Llama 3 Tulu V2 8B RM, we also calculate the win rates for the best coefficient of KL and $\chi^2$ regularization. We use AlpacaEval (Dubois et al., 2023) with GPT4o-mini to compute the win rate between the RLHF policy and the SFT policy. We find that the median win rate for $\chi^2$ divergence is higher, and it is also more consistent across random seeds. In contrast, using KL divergence leads to reward hacking for one seed and a lower median win rate.

| Divergence | Coefficient | Median win rate | Win rate range |
|---|---|---|---|
| $\chi^2$ | 0.0008 | **52.83** | 51.98 – 53.98 |
| KL | 0.025 | 51.50 | 11.93 – 53.75 |

Table 4: Win rates for RLHF-trained models using KL divergence vs. $\chi^2$ divergence. The median win rate and range of win rates are reported across five seeds.

## E.2  RESULTS FOR ALL REGULARIZATION COEFFICIENTS

Here, we present the results of training with AD and OM regularization using $\chi^2$ and KL divergence across all regularization coefficients. Each table shows the median and the standard deviation of the true rewards achieved by the learned policy across 5 random seeds.

| $\chi^2$ divergence | | | |
|---|---|---|---|
| **Coefficient** | **AD** | **State OM** | **State-action OM** |
| 0.000002 | -60.79 $\pm$ 8.40 | -61.04 $\pm$ 2.20 | -59.11 $\pm$ 4.53 |
| 0.000004 | -62.01 $\pm$ 8.58 | -61.85 $\pm$ 2.22 | -58.21 $\pm$ 5.52 |
| 0.00001 | -60.43 $\pm$ 6.61 | -59.82 $\pm$ 2.85 | -54.35 $\pm$ 2.81 |
| 0.00002 | -62.01 $\pm$ 4.54 | -56.53 $\pm$ 10.66 | -1.35 $\pm$ 30.28 |
| 0.00004 | -50.84 $\pm$ 6.28 | -42.24 $\pm$ 22.01 | -1.15 $\pm$ 0.06 |
| 0.0001 | -1.29 $\pm$ 0.12 | -2.18 $\pm$ 0.42 | -1.38 $\pm$ 0.29 |
| 0.0002 | -1.70 $\pm$ 0.12 | -2.46 $\pm$ 0.65 | -1.85 $\pm$ 0.29 |

| KL divergence | | | |
|---|---|---|---|
| **Coefficient** | **AD** | **State OM** | **State-action OM** |
| 0.000001 | -56.20 $\pm$ 4.02 | -61.59 $\pm$ 2.72 | -61.18 $\pm$ 2.87 |
| 0.0000025 | -59.58 $\pm$ 4.84 | -54.59 $\pm$ 3.16 | -59.59 $\pm$ 2.36 |
| 0.000005 | -57.24 $\pm$ 2.62 | -59.03 $\pm$ 5.41 | -61.24 $\pm$ 2.01 |
| 0.00001 | -54.84 $\pm$ 3.15 | -57.62 $\pm$ 3.13 | -58.85 $\pm$ 2.73 |
| 0.000025 | -55.10 $\pm$ 2.64 | -59.32 $\pm$ 1.37 | -56.86 $\pm$ 5.48 |
| 0.00005 | -49.99 $\pm$ 4.04 | -59.96 $\pm$ 1.65 | -53.39 $\pm$ 23.77 |
| 0.0001 | -45.72 $\pm$ 9.02 | -1.34 $\pm$ 25.30 | -1.25 $\pm$ 0.07 |
| 0.00025 | -1.33 $\pm$ 0.05 | -1.47 $\pm$ 0.20 | -1.51 $\pm$ 0.10 |
| 0.0005 | -1.52 $\pm$ 0.04 | -1.76 $\pm$ 0.22 | -1.99 $\pm$ 0.35 |
| 0.001 | -1.73 $\pm$ 0.07 | -1.76 $\pm$ 0.26 | -2.30 $\pm$ 1.14 |
| 0.0025 | -1.98 $\pm$ 0.07 | -1.76 $\pm$ 0.51 | -1.94 $\pm$ 0.30 |
| 0.005 | -2.15 $\pm$ 0.05 | -1.90 $\pm$ 0.83 | -2.08 $\pm$ 0.61 |
| 0.01 | -2.11 $\pm$ 0.05 | -2.12 $\pm$ 1.00 | -2.14 $\pm$ 0.56 |

Table 5: All traffic control results ($\times 10^3$).

$\chi^2$ divergence

| Coefficient | AD | State OM | State-action OM |
|---|---|---|---|
| 0.0008 | $-17.60 \pm 1.78$ | $-12.59 \pm 1.63$ | $-27.16 \pm 12.83$ |
| 0.0016 | $-36.16 \pm 3.25$ | $-11.25 \pm 10.08$ | $-11.17 \pm 0.19$ |
| 0.004 | $-34.16 \pm 12.90$ | $-10.88 \pm 0.91$ | $-11.65 \pm 2.12$ |
| 0.008 | $-12.45 \pm 0.25$ | $-10.99 \pm 3.13$ | $-12.02 \pm 0.18$ |
| 0.016 | $-12.31 \pm 0.08$ | $-10.68 \pm 0.17$ | $-12.18 \pm 0.46$ |
| 0.04 | $-12.29 \pm 0.05$ | $-10.78 \pm 0.12$ | $-12.34 \pm 0.10$ |
| 0.08 | $-12.39 \pm 0.04$ | $-10.73 \pm 0.84$ | $-12.20 \pm 0.11$ |

KL divergence

| Coefficient | AD | State OM | State-action OM |
|---|---|---|---|
| 0.00006 | $-21.23 \pm 9.74$ | $-33.59 \pm 9.89$ | $-30.96 \pm 22.42$ |
| 0.00012 | $-30.39 \pm 22.16$ | $-41.96 \pm 12.70$ | $-19.67 \pm 6.43$ |
| 0.0003 | $-23.10 \pm 5.63$ | $-35.29 \pm 10.32$ | $-27.56 \pm 7.50$ |
| 0.0006 | $-21.85 \pm 19.03$ | $-34.56 \pm 11.97$ | $-22.40 \pm 9.84$ |
| 0.0012 | $-25.17 \pm 10.24$ | $-31.28 \pm 7.83$ | $-31.77 \pm 6.01$ |
| 0.003 | $-23.51 \pm 6.91$ | $-35.76 \pm 10.53$ | $-23.90 \pm 12.81$ |
| 0.006 | $-12.26 \pm 11.82$ | $-58.08 \pm 46.98$ | $-29.42 \pm 25.37$ |
| 0.012 | $-12.30 \pm 9.27$ | $-10.60 \pm 0.87$ | $-11.88 \pm 0.81$ |
| 0.03 | $-12.28 \pm 0.14$ | $-11.03 \pm 6.86$ | $-11.73 \pm 0.21$ |
| 0.06 | $-12.20 \pm 0.07$ | $-10.71 \pm 0.18$ | $-12.23 \pm 14.25$ |
| 0.12 | $-12.33 \pm 0.04$ | $-10.24 \pm 0.61$ | $-12.09 \pm 0.38$ |
| 0.3 | $-12.35 \pm 0.04$ | $-11.02 \pm 0.57$ | $-12.11 \pm 0.25$ |
| 0.6 | $-12.40 \pm 0.04$ | $-10.61 \pm 0.36$ | $-12.11 \pm 0.28$ |
| 1.2 | $-12.33 \pm 0.03$ | $-10.50 \pm 0.26$ | $-12.02 \pm 0.28$ |

Table 6: All pandemic mitigation results.

| $\chi^2$ divergence | | | |
|---|---|---|---|
| **Coefficient** | **AD** | **State OM** | **State-action OM** |
| 0.0005 | $-580.7 \pm 73.8$ | $-484.2 \pm 56.6$ | $-164.0 \pm 3.5$ |
| 0.001 | $-94.2 \pm 14.0$ | $-263.9 \pm 19.2$ | $-127.0 \pm 5.9$ |
| 0.0025 | $-97.1 \pm 8.2$ | $-146.4 \pm 18.1$ | $-93.3 \pm 9.3$ |
| 0.005 | $-76.6 \pm 10.5$ | $-109.7 \pm 10.3$ | $-55.2 \pm 0.7$ |
| 0.01 | $-84.7 \pm 8.5$ | $-72.8 \pm 8.8$ | $-47.5 \pm 0.6$ |
| 0.025 | $-85.6 \pm 7.9$ | $-54.3 \pm 1.3$ | $-50.9 \pm 2.3$ |
| 0.05 | $-74.8 \pm 13.1$ | $-57.1 \pm 3.5$ | $-113.3 \pm 32.8$ |

| KL divergence | | | |
|---|---|---|---|
| **Coefficient** | **AD** | **State OM** | **State-action OM** |
| 0.00003 | $-598.4 \pm 39.7$ | $-604.1 \pm 10.7$ | $-583.8 \pm 61.2$ |
| 0.00006 | $-600.6 \pm 12.4$ | $-589.0 \pm 260.9$ | $-594.7 \pm 3.3$ |
| 0.00015 | $-592.3 \pm 51.3$ | $-607.9 \pm 11.1$ | $-577.1 \pm 11.2$ |
| 0.0003 | $-593.6 \pm 6.0$ | $-592.0 \pm 29.3$ | $-497.9 \pm 11.2$ |
| 0.0006 | $-590.0 \pm 7.5$ | $-593.1 \pm 5.4$ | $-364.3 \pm 5.8$ |
| 0.0015 | $-459.9 \pm 114.1$ | $-511.1 \pm 21.1$ | $-181.6 \pm 7.5$ |
| 0.003 | $-270.0 \pm 39.7$ | $-332.3 \pm 41.0$ | $-101.2 \pm 5.0$ |
| 0.006 | $-154.5 \pm 5.5$ | $-158.7 \pm 28.8$ | $-61.9 \pm 5.2$ |
| 0.015 | $-84.1 \pm 6.8$ | $-82.9 \pm 5.6$ | $-48.9 \pm 0.5$ |
| 0.03 | $-98.3 \pm 8.4$ | $-58.4 \pm 3.8$ | $-49.6 \pm 1.2$ |
| 0.06 | $-88.6 \pm 12.8$ | $-59.0 \pm 7.2$ | $-78.3 \pm 10.2$ |
| 0.15 | $-82.1 \pm 11.6$ | $-75.9 \pm 4.6$ | $-106.6 \pm 19.6$ |
| 0.3 | $-73.4 \pm 9.2$ | $-98.1 \pm 16.1$ | $-127.3 \pm 24.7$ |
| 0.6 | $-88.6 \pm 5.6$ | $-112.5 \pm 16.6$ | $-118.5 \pm 10.7$ |

Table 7: All glucose monitoring results ($\times 10^3$).

| **Coefficient** | **AD $\chi^2$** | **AD KL** |
|---|---|---|
| 0.00008 | $9.20 \pm 0.68$ | $8.80 \pm 2.24$ |
| 0.00025 | $14.05 \pm 3.27$ | $9.48 \pm 1.02$ |
| 0.0008 | $16.94 \pm 0.07$ | $8.84 \pm 0.42$ |
| 0.0025 | $16.84 \pm 0.08$ | $14.22 \pm 2.81$ |
| 0.008 | $16.71 \pm 0.11$ | $12.73 \pm 2.75$ |
| 0.025 | $16.59 \pm 0.11$ | $16.81 \pm 0.27$ |
| 0.08 | $16.43 \pm 0.07$ | $16.52 \pm 0.08$ |
| 0.25 | $16.22 \pm 0.10$ | $16.33 \pm 0.04$ |
| 0.8 | $16.13 \pm 0.10$ | $16.25 \pm 0.13$ |

Table 8: All results for RLHF.

| $\chi^2$ divergence | | | |
|---|---|---|---|
| **Coefficient** | **AD** | **State OM** | **State-action OM** |
| 0.0005 | $2.65 \pm 0.67$ | $0.64 \pm 0.19$ | $0.53 \pm 0.29$ |
| 0.001 | $3.87 \pm 0.25$ | $0.65 \pm 4.63$ | $0.60 \pm 0.16$ |
| 0.0025 | $6.24 \pm 0.10$ | $9.06 \pm 0.17$ | $9.04 \pm 4.71$ |
| 0.005 | $6.17 \pm 0.03$ | $9.06 \pm 0.11$ | $9.16 \pm 0.09$ |
| 0.01 | $6.16 \pm 0.04$ | $9.07 \pm 0.07$ | $9.17 \pm 0.12$ |
| 0.025 | $6.19 \pm 0.04$ | $8.64 \pm 0.12$ | $8.61 \pm 0.18$ |
| 0.05 | $6.14 \pm 0.00$ | $7.95 \pm 0.05$ | $7.99 \pm 0.22$ |

| KL divergence | | | |
|---|---|---|---|
| **Coefficient** | **AD** | **State OM** | **State-action OM** |
| 0.0008 | $2.52 \pm 0.18$ | $2.32 \pm 0.86$ | $2.31 \pm 0.08$ |
| 0.0016 | $2.98 \pm 0.33$ | $2.31 \pm 0.07$ | $1.11 \pm 0.89$ |
| 0.004 | $4.59 \pm 0.19$ | $2.23 \pm 0.06$ | $2.01 \pm 0.23$ |
| 0.008 | $6.10 \pm 0.15$ | $1.30 \pm 0.19$ | $1.25 \pm 0.32$ |
| 0.016 | $6.33 \pm 0.13$ | $0.82 \pm 0.22$ | $0.84 \pm 0.21$ |
| 0.04 | $6.26 \pm 0.05$ | $1.17 \pm 2.92$ | $1.81 \pm 0.31$ |
| 0.08 | $6.26 \pm 0.04$ | $7.62 \pm 0.06$ | $7.32 \pm 0.28$ |
| 0.16 | $6.21 \pm 0.05$ | $7.20 \pm 0.12$ | $7.12 \pm 0.11$ |
| 0.4 | $6.16 \pm 0.03$ | $6.89 \pm 0.14$ | $6.84 \pm 0.19$ |
| 0.8 | $6.19 \pm 0.03$ | $7.07 \pm 0.12$ | $6.86 \pm 0.19$ |
| 1.6 | $6.14 \pm 0.04$ | $6.90 \pm 0.13$ | $6.61 \pm 0.31$ |
| 4 | $6.13 \pm 0.03$ | $6.80 \pm 0.16$ | $6.79 \pm 0.12$ |
| 8 | $6.13 \pm 0.01$ | $6.81 \pm 0.28$ | $6.80 \pm 0.06$ |
| 16 | $6.13 \pm 0.00$ | $6.94 \pm 0.10$ | $6.83 \pm 0.25$ |

Table 9: All results for the tomato-watering gridworld.

### E.3 ABLATIONS

Here, we present the results of two ablations of ORPO. For each of the non-RLHF environments, we fix the optimal coefficient for state-action OM $\chi^2$ regularization and modify other hyperparameters. We do not ablate the RLHF experiments because RLHF is a contextual bandit (Appendix A.3) and so it isn't actually necessary to run ORPO for RLHF. The results are shown in Table 10 below.

**Order of training policy and discriminator networks.** First, we experiment with modifying ORPO to train the discriminator *after* the policy. In Algorithm 1, the discriminator $\hat{d}_\phi$ is optimized, then the rewards are updated with the discriminator outputs, and then the policy is trained with the updated rewards. An alternative is to wait to train the discriminator until after the policy has been updated, i.e., put Line 4 after Line 7. We experimented with this and found that in in most environments there is not too much difference between the two orders, although training the discriminator first gives slightly better results. However, in the pandemic mitigation environment, we found that it training the discriminator second gave results with much higher variance. This suggests that it is best to train the discriminator before augmenting the rewards to train the policy.

**Discriminator reward clipping.** Second, we experiment with modifying the discriminator reward clipping parameter $\delta$ of ORPO. We found that removing the clipping parameter entirely led to NaN errors and training could not complete, so we do not report those results. However, to test sensitivity to this parameter, we tried training with a clipping parameter $10\times$ larger and $10\times$ smaller in each environment. We found that the results did not vary by much across different clipping parameters. This suggests that ORPO is relatively robust to the hyperparameter, so it does not need to be tuned precisely.

| Method | Environment | | | |
|---|---|---|---|---|
| | Traffic control $(\times 10^3)$ | Pandemic mitigation | Glucose monitoring $(\times 10^3)$ | AI safety gridworld |
| Default parameters | $-1.15 \pm 0.06$ | $-10.68 \pm 0.15.$ | $-47.5 \pm 0.6$ | $9.17 \pm 0.12$ |
| Train policy before discriminator | $-1.24 \pm 0.07$ | $-11.51 \pm 0.40$ | $-48.2 \pm 0.7$ | $8.97 \pm 0.13$ |
| Discriminator reward clipping $\times 0.1$ | $-1.18 \pm 0.14$ | $-10.77 \pm 0.16$ | $-47.7 \pm 0.8$ | $9.21 \pm 0.18$ |
| Discriminator reward clipping $\times 10$ | $-1.24 \pm 0.03$ | $-10.73 \pm 0.37$ | $-47.9 \pm 0.7$ | $9.20 \pm 0.14$ |

Table 10: Results of our ablations of ORPO. We report the median and standard deviation of the true reward across five random seeds for the four non-RLHF environments. The top row shows results using each environment's most effective form of occupancy measure regularization: state-action OM for traffic, glucose, and tomato and state OM for pandemic, with the optimal $\chi^2$ divergence coefficients. The other rows show ablations with the same coefficient. We find that ORPO is mostly robust to different hyperparameters but that it is probably best to train the discriminator network before the policy network.

