# OpenReview forum: "Correlated Proxies: A New Definition and Improved Mitigation for Reward Hacking"
_ICLR.cc/2025/Conference — ICLR 2025 Spotlight_

### Official Review · Reviewer_Lc7U · 2024-10-27

**Soundness:** 3
**Presentation:** 4
**Contribution:** 3
**Rating:** 8
**Confidence:** 3

**Summary:**

The paper studies reward hacking, including in the context of reinforcement learning from human feedback. It proposes a new definition of what constitutes reward hacking and shows that a specific form of regularization can help with preventing reward hacking. In the context of RLHF, the results suggest a different regularization scheme than a standard KL penalty. The authors further argue that this type of regularization better mitigates reward hacking in practice.

**Strengths:**

The paper is well-written, enjoyable to read, and relatively easy to follow. It provides an interesting perspective on reward hacking through the lens of regularized RL objectives. The approach is based on a new definition of reward hacking, which assumes access to a base policy-a reference used to determine whether a proxy reward is hackable by a learning agent. This definition is intuitive, although I have some questions and comments about its expressiveness.

I find the connection to the KL-regularized RLHF objective, and the corresponding result (Theorem 3.1), particularly interesting. To the best of my knowledge, this result is novel. The experimental results are extensive and, in my opinion, demonstrate the effectiveness of the proposed approach.

**Weaknesses:**

Regularization to a base policy in RLHF for language models is somewhat natural, as we aim to preserve the language generation capabilities of our policy. However, as shown in Table 1, it can also lead to considerable performance loss in some environments (e.g., Pandemic Mitigation), where the base policy does not perform well. I wonder how preference-based RL (without regularization) would fare in such environments. It seems that the proposed approach is somewhat constrained, as it requires humans to provide a reliable reference point (base policy), which may be challenging in complex environments.

While I am generally positive about this work, I think I didn’t fully understand how it compares to some other well-known approaches to mitigating similar problems (e.g., techniques based on inverse reward design or risk-averse RL). I understand that these approaches do not necessarily address the same problem, but the experimental setup does not seem to include baselines that would illustrate the importance of Definition 4.2. For example, since $\pi_{base}$ is given, why isn’t IRL an adequate baseline?

The related work section covers most of the important references, but a more comprehensive literature review on RLHF could be beneficial. For example, the paper mentions:

> Our regularized objective in (5) differs in two main ways from the KL regularization used in RLHF. First, our results suggest optimizing the occupancy measure (OM) divergence between policies, whereas RLHF uses the action distribution (AD) divergence.

however, it does not explain how this connects to recent results that propose OM-based regularizers. For example, it would be useful if the authors could comment on the MDP formulation of RLHF from

> Nika et al., Reward Model Learning vs. Direct Policy Optimization: A Comparative Analysis of Learning from Human Preferences, ICML 2024.

which in Section 7.2 introduces the KL-regularized RHF objective based on occupancy measures.

One potential drawback of the proposed approach is that it is based on estimating occupancy measures, which may be challenging to estimate for high-dimensional state spaces. It would be helpful if the authors could comment on the level of difficulty of the current environments for an RL agent. Furthermore, how accurate are the occupancy measures obtained in the experiments? It seems that it should be possible to estimate this for some of the environments.

Some minor details in the experimental setup were not clear to me. The appendix states:

> We primarily tuned the hyperparameters listed below in order to ensure that the proxy reward would be properly optimized and reward hacking would occur without regularization.

Does this mean that we typically don't observe reward hacking in these environments?

**Questions:**

Please see my comments and questions *Weaknesses*. I'd appreciate if the authors could respond to these comments and questions.

---

> ### Author Response · Authors · 2024-11-20
> **Rebuttal**
>
> We thank the reviewer for their thorough review and many helpful suggestions. We are glad that they found the paper to be "well-written", and that it "provides an interesting perspective on reward hacking." Below, we have addressed individual points in the review:
>
>   * **Comparison to preference-based RL:** The reviewer suggests that in some environments preference-based RL (PBRL) may be able to outperform regularized policy optimization. We believe that a comparison to PBRL is out-of-scope for our problem setting, since we are interested in how to optimize a misspecified proxy reward *without further input*. In contrast, PBRL generally requires multiple rounds of feedback from a human annotator during learning. However, regularized policy optimization is often a key subcomponent of PBRL, since once a batch of preference data is collected, it is often optimized using a regularized objective, as in RLHF. Thus, we believe that our results may be useful in improving PBRL algorithms.
>   * **Comparison to inverse reward learning:** The reviewer also suggests comparing our algorithm to inverse reinforcement learning (IRL). However, we note that IRL is generally used for producing a reward function that can be used to *imitate* demonstrations, and thus, if we applied IRL to the base policy, we would simply recover equal performance to it. In contrast, we aim to *improve* performance over the base policy using a reward function which is misspecified but correlates with the true objective.
>   * **OM-based regularization for RLHF:** We appreciate the reviewer bringing this important reference to our attention. Nika et al. [1] do consider an objective with log-linear occupancy measure regularization. However, their analysis is limited to deterministic MDPs, and they do not evaluate their approach empirically. In contrast, we show both in theory and in our experiments, that our approach to preventing reward hacking can succeed in general stochastic environments. We have added a reference to [1] in our related work.
>   * **Estimating occupancy measures in high-dimensional state spaces:** Our approach to estimating occupancy measure divergence in ORPO is based on using a learned discriminator, which has succeeded in many similar applications (e.g., GANs [2] and GAIL [3]). We test ORPO in environments with state space dimensions ranging from 1 (glucose monitoring) to 50 (traffic control), showing that it is effective in high-dimensional spaces. Furthermore, we have added Figure 6 to Appendix B showing that the learned discriminator for the glucose monitoring environment is close to the exact log ratio of the occupancy measures, validating that our approximation of occupancy measure divergence is accurate.
>   * **Tuning of hyperparameters:** We found that in some environments, if the RL hyperparameters were not tuned, PPO would simply fail to optimize the proxy reward. In these cases, reward hacking did not occur, while PPO also failed to improve the true reward. To study various techniques to prevent reward hacking, we tuned PPO such that it can consistently optimize the proxy reward, leading to reward hacking in all environments.
>
> [1] Nika et al. Reward Model Learning vs. Direct Policy Optimization: A Comparative Analysis of Learning from Human Preferences. ICML 2024.
>
> [2] Goodfellow et al. Generative Adversarial Nets. NIPS 2014.
>
> [3] Ho and Ermon. Generative Adversarial Imitation Learning. NIPS 2016.

---

> > ### Comment · Reviewer_Lc7U · 2024-11-26
> >
> > Thank you for your response. It addresses my concerns, and I will update my score accordingly.

---

### Official Review · Reviewer_icTG · 2024-11-03

**Soundness:** 3
**Presentation:** 2
**Contribution:** 3
**Rating:** 8
**Confidence:** 3

**Summary:**

This paper critically formalizes a notion of exploitability in RL-based
fine-tuning when the reward function is difficult or impossible to
specify exactly, and is instead replaced by a "proxy" reward. It argues
that reward hacking is characterized by the proxy reward admitting
optimal policies that are in fact *not* aligned with our desired
behavior, and their characterization is validated with several intuitive
case studies. Moreover, under this characterization, the authors propose
a RL fine-tuning algorithm that prevents reward hacking by regularizing
the *occupancy measure* of the fine-tuned policy towards a base policy
with respect to $\chi^2$ divergence, under weaker assumptions on the
proxy reward than existing results. Moreover, the paper introduces a
practical implementation using techniques from inverse RL (particularly
GAIL), and demonstrate that their method performs better than baselines
across several fine-tuning tasks.

**Strengths:**

This paper studies a very hot topic in the field, and in my opinion,
provides some very interesting insight. Most notably, I really enjoyed
the following entangled results in the paper:

1.  Even with a proxy reward that is highly correlated with the ground
    truth reward, reward hacking is possible;
2.  But under such correlated rewards, it is (probably) possible to
    prevent reward hacking by regularizing the fine-tuned policie's
    occupancy measure (wrt $\chi^2$).

It is also nice that their proposed framework fits nicely with that of
IRL, allowing the authors to leverage performant IRL algorithms for RL
fine-tuning.

**Weaknesses:**

My main concerns with the paper regard lack of clarity in several
respects. I will outline these below, and further examples are given in
"Minor Issues" and "Questions".

1.  I find the narrative of the paper to be a little misguided. The
    paper goes on to try to essentially axiomatize reward hacking,
    leading to many fairly philosophical or imprecise heuristic
    discussions (e.g. "it's not reward hacking if the proxy is just
    bad!"). I think this is, in a sense, actually overcomplicating
    things. Under the hood, I think there is a very clear and
    uncontroversial message here:
    1.  When fine-tuning, we want to learn a policy that is better than
        some base policy at maximizing some ground truth reward (this is
        hard to argue with);
    2.  Proxy rewards that are highly correlated to the ground truth
        reward are still "hackable", in the sense that policy
        optimization with such proxies prohibits improvement over the
        base policy (the authors show this);
    3.  But hackable correlated proxies are not useless—it is possible
        to successfully fine-tune a base policy with such a proxy using
        $\chi^2$ regularization (the authors basically show this, see
        Questions).
2.  While interesting, I think some of the theoretical results are
    actually fairly weak (see Questions). For instance, while the
    theoretical results suggest regularization of the fine-tuned
    policy's occupancy measure, they do not actually suggest that this
    should be an improvement over action-distribution regularization.
    Moreover, it is not actually clear that the lower bound of Theorem
    5.1, which is the main theorem of the paper, can actually be
    positive (in other words, Theorem 5.1 does not explicitly show that
    fine-tuning with occupancy measure regularization can actually lead
    to improved alignment). I suspect it is possible for the authors to
    correct these, and that would strengthen the paper.
3.  Overall, there are several issues with writing, such as discussion
    of related work, redundancy in intro/background sections, and some
    terminology, which I outlined below.

My score is mostly a reflection of weakness 2 here. I trust that the authors can fix weaknesses 1 and 3 fairly easily. Should the authors address weakness 2 (as well as the questions below), I would be happy to increase my score.

## Minor Issues

Stiennon et al. (2020) is definitely not the first example of using KL
regularization to an "offline policy". This is very commonly done in all
of maximum entropy / entropy-regularized reinforcement learning, dating
back at least fifteen years before this reference.

With regard to prior work in offline RL that enforce that the learned
policy does not veer too far away from the data distribution, the work
of \[2\] is the most appropriate reference (see list of references in
Questions section below).

The paper starts off much too slow. The first four pages are fairly
redundant, the isssue of formalizing the notion of reward hacking is
addressed several times, and only very heuristic high-level ideas are
suggested for addressing the issue. I believe this pages can be
condensed a lot, which would make the paper easier to read.

The section about the desiderata for a definition of reward hacking can
be clarified a lot. In particular, it is not clear to me exactly what
the desiderata are, they should be stated very explicitly (especially
since the paper later refers to them in some order, e.g. "the second
desideratum").

At the bottom of page 5, $\sigma_{\tilde{R}}$ and $\sigma_R$ are really
variances, not standard deviations (so they should probably be written
as $\sigma_{\bullet}^2$).

In definition 4.2, there is a missing word — "*reward hacking* occurs
when an optimal policy $\tilde{R}$ **lower** true reward…".

In equation (8), you use $\phi$ on both sides of the equation (you
should use e.g. $\phi'$ as the variable being minimized).

While definition 4.2 is interesting, I don't think "reward hacking" is
the right name for the phenomenon being described in this definition.
Particularly, this definition presents a property on a proxy reward, but
not on the process of fine-tuning a policy with this proxy. Indeed, one
of the contributions presented later in the work is an algorithm that
prevents reward hacking under proxies satisfying definition 4.2! Rather,
this definition presents conditions under which reward hacking *may*
occur. As such, I believe "reward exploitability" is an example of a
more appropriate name: standard policy optimizations *can* exploit/hack
the reward function if you're not careful.

On line 313, "polulation" should be "population".

**Questions:**

In the introduction, you claim that the formalization of a "good proxy"
has been elusive. Hasn't this issue been addressed by the EPIC distance
\[1\]? What does EPIC leave to be desired?

Is there a mistake in the Table 1 cell for traffic control with state
occupancy KL? Particularly, the standard deviation of $22.6$ is
alarming.

Theorem 5.1 is nice in the sense that it tells us that occupancy measure
regularization (particularly w.r.t $\chi^2$) can prevent reward hacking
even when the proxy is correlated with the true reward. However, there
is no claim about whether or not reward hacking can (at least in
principle) be prevented with action distribution regularization. So,
naturally, I'm curious whether or not a similar result can be achieved
with action distribution regularization. The empirical results suggest
not, but perhaps we just haven't found the right way to do it yet. Do
you think it's impossible?

In the experiments, RLHF is treated as a contextual bandit problem, in
which case there is no difference between occupancy measure
regularization and action distribution regularization (because state
occupancy is independent of actions). However, especially given the
autoregressive nature of many existing LLMs, RLHF can also be viewed as
a sequential problem where the policy sequentially chooses individual
tokens, and a reward is presented after the last token is chosen. How
would OM under this formulation of RLHF compare to AD in the "atomic"
contextual bandit case?

Theorem 5.1 looks nice because intuitovely, it tells us that by
controlling the $\chi^2$ between the learned policy and the base policy,
we can prevent reward hacking. But(!), it is actually not entirely clear
to me if this lower bound admits the possibility that we can actually
*improve* upon the base policy using this regularization. Clearly, as
the $\chi^2$ divergence tends to $0$, the lower bound tends to $0$. But
can the lower bound ever be greater than $0$?

## References

1.  Gleave, Dennis & Legg et al. (2021) Quantifying Differences in
    Reward Functions, arXiv.
2.  Fujimoto, Meger & Precup (2019) Off-Policy Deep Reinforcement
    Learning Without Exploration, ICML.

---

> ### Author Response · Authors · 2024-11-20
> **Rebuttal**
>
> We thank the reviewer for their thorough review. We are glad they found our paper "provides some very interesting insight." We have responded to individual points in the review below:
>   * **Can the lower bound in Theorem 5.1 be optimized?** We agree that it would be ideal if we could guarantee that the lower bound on the true reward in Theorem 5.1 can be increased above 0. However, it turns out to be quite difficult to prove this in general; we have added Lemma A.3 in Appendix A.1.3 showing that there are MDPs for any correlation coefficient $r$ where it is impossible to increase the true reward by optimizing the lower bound. This holds true even when there is a policy that improves both true and proxy reward. However, the advantage of the lower bound in Theorem 5.1 is that it always allows safe optimization of the proxy reward: even if it is not possible to increase the lower bound above zero, optimizing the objective will at least prevent reward hacking. If we are able to increase the objective, then we know that the true reward must have increased too. Our empirical results suggest that in many realistic environments it *is* possible to increase the lower bound in Theorem 5.1.
>   * **"There is no claim about whether or not reward hacking can (at least in principle) be prevented with action distribution regularization":** We have added Theorem A.5 to Appendix A.2 (and referenced it from the main text) showing that action distribution regularization cannot achieve the same guarantees as occupancy measure regularization. Theorem A.5 is very general: it shows that regularizing using *any* $f$-divergence between action distributions scaled using *any* function cannot prevent reward hacking.
>   * **EPIC distance:** The EPIC distance could serve as a formalization of what makes a "good" proxy. However, there are multiple barriers in applying it to the environments we study. First, it only applies to reward functions of the form $R(s, a, s')$ that depend on the next state as well as the current state and action; none of the environments we explore have this type of reward function. Furthermore, it requires *independent* state and action distributions to use for computing the canonically shaped reward, whereas we use occupancy measures where the state and action distributions are not independent. For these reasons, we do not use it in our study. However, we have added a reference to the related work.
>   * **RLHF as a contextual bandit problem:** We agree that RLHF can also be formulated with each generated token representing a separate action. We have added Lemma A.7 in Appendix A.3 showing that in this case, action distribution and occupancy measure KL divergence are still equivalent.
>   * **Table 1 cell for traffic control with state occupancy KL:** This is not a typo; RL is quite unstable in the traffic environment with state-only KL occupancy regularization. This is likely because the reward function in the traffic environment depends on both the state and action, so our theory suggests state-only regularization will not work--we only include it for completeness. As shown in Table 4 in the Appendix, a slightly larger regularization coefficient achieves true reward of -1.47 ± 0.20, which is a slightly lower median but with much less variance.
>   * **First example of KL regularization to an "offline policy":** We agree that maximum entropy RL can be understood as KL regularization to the static uniform policy. However, to our knowledge this is not a common tool used to prevent reward hacking. Instead, it usually aids in exploration or is used to model human behavior. We have clarified in the related work that Stiennon et al. [1] are the first to consider KL regularization to a fixed policy *in the context of reward hacking*.
>   * **Related work in offline RL:** We have added the Fujimoto et al. reference to our related work, and we additionally reference several other works on offline RL there.
>   * **Definition 4.2 ("reward hacking"):** We agree that this may not be the best term for the phenomenon. Based on the feedback in this review and that of Reviewer qUra, we have updated the definition to instead refer to the reward function as "hackable" if the condition is met.
>   * **Narrative of the paper:** We appreciate the reviewer's suggestions for improving the narrative of the paper. Our choice of narrative was based on previous rounds of reviewing in which we received comments like "it is not entirely clear what the authors mean by the term reward hacking" and "this work lacks a global view and theory on the problem." Because of these comments, we believe that the extended introduction and discussion of the definition of reward hacking is helpful for a broad audience to understand and appreciate the paper.
>   * **Other notes:** We have fixed minor issues in an updated version of the paper that we uploaded to OpenReview.
>
> [1] Stiennon et al. Learning to summarize from human feedback. NeurIPS 2020.

---

> > ### Comment · Reviewer_icTG · 2024-11-26
> >
> > Thanks a lot for the detailed response.
> >
> > - **Can the lower bound in Theorem 5.1 be optimized?**: I appreciate this discussion, but it isn't entirely answering the question I had. You argue that it is generally difficult to guarantee that the lower bound is positive, but I am curious to know if the lower bound *can* be positive. That is, does there exist an MDP and an r-correlated proxy where the lower bound is positive?
> > - **"There is no claim about whether or not reward hacking can (at least in principle) be prevented with action distribution regularization"**: Thanks a lot, very neat!
> > - **Reward hacking --> Hackable reward**: Thanks!
> >
> > Most of my other concerns have been addressed. I am still curious about the first point mentioned in this comment, but I'll increase my score to 6 in light of the revised draft and discussion.

---

> ### Author Response · Authors · 2024-11-26
>
> Thank you for responding to our rebuttal! Regarding whether the lower bound in Theorem 5.1 can be positive, we have added Lemma A.4 to Appendix A.1.3. This lemma shows that for *any* correlation coefficient $r \in (0, 1)$, there is an MDP where the lower bound in Theorem 5.1 can be positive, and additionally, in this MDP optimizing the lower bound leads to a policy with the optimal true reward. We hope this addresses your concern.

---

> > ### Comment · Reviewer_icTG · 2024-11-28
> >
> > Thanks again for the hard work, I think Lemma A.4 is a nice addition and I believe it should go in the main text. Appendix A in particular now greatly clears up the story for me, and I will raise my score.

---

### Official Review · Reviewer_4snh · 2024-11-03

**Soundness:** 4
**Presentation:** 4
**Contribution:** 4
**Rating:** 8
**Confidence:** 4

**Summary:**

The authors provide a new definition of reward hacking based on a new definition of correlated policies. Reward hacking is a phenomenon that occurs when optimizing a proxy reward for a true reward function yields worse performance over the true reward. The definition of correlated captures proxy reward functions that match the true reward function in visited state action pairs by a base policy. In addition, the authors propose occupancy measure regularization with respect to a base policy as a method to overcome reward hacking. They show theoretically and empirically that the proposed method mitigates reward hacking.

**Strengths:**

- The paper is very well written with an astounding flow of information.
- The authors provide a lot of great intuitive examples for the arguments they make.
- The authors provide a wide range of experiments that show both the soundness and robustness of the claims.

**Weaknesses:**

- In Theorem 5.1, there is the assumption that the state-occupancy measure of the policy is absolutely continuous with respect to the base policy. However, during training, this is not something that can be guaranteed. I suggest adding an extra discussion on how regularization also pushes toward the learned policy satisfying this assumption.

**Questions:**

- The definition of correlated rewards allows for the correlated reward to deviate arbitrarily in states with zero occupancy by the base policy.  Doesn't this allow for policies that can be dangerous to still be correlated with the true reward? For example, if the base policy avoids a state with a large negative true reward but the proxy reward function assigns a very high positive reward for that state. Wouldn't this reward incentivize policies that try to visit this state and get a large negative true reward? In addition, wouldn't these kinds of proxy create policies that break the assumption in Theorem 5.1?

---

> ### Author Response · Authors · 2024-11-20
> **Rebuttal**
>
> We thank the reviewer for their review and are glad that they found the paper "very well written" with "a wide range of experiments that show both the soundness and robustness of the claims." Below, we have addressed the weaknesses and questions raised in the review:
>
> **Assumption that the state-occupancy measure of the policy is absolutely continuous with respect to the base policy:** Theoretically, we need this assumption because otherwise the $\chi^2$ divergence is undefined (there is a division by zero in the definition). However, in practice, we believe it is not a significant problem for a few reasons. First, in all experiments we initialize training with the base policy. Thus, at the beginning of training, the occupancy measure of the RL policy is clearly absolutely continuous w.r.t. that of the base policy. Second, during training, the $\chi^2$ penalty strongly disincentivizes reaching state-action pairs that have very low probability under the base policy's occupancy measure, since the $\chi^2$ divergence increases with $1 / \mu_{\pi\_\text{base}}(s, a)$. Therefore, the penalty is likely to push policy optimization away from areas of the state-action space where $\mu_{\pi\_\text{base}}(s, a)$ is very small, making it unlikely that the RL policy $\pi$ will reach states where $\mu_{\pi\_\text{base}}(s, a) = 0$.
>
> Third, in practice we do not exactly calculate the $\chi^2$ divergence but approximate it with a discriminator. Suppose the RL policy $\pi$ does reach a state with zero probability under the base policy. In this case, while the exact $\chi^2$ divergence would be undefined, in practice the discriminator will assign an extremely high (but not infinite) score to the state, strongly penalizing it and causing RL to avoid it.
>
> Thus, since the RL policy begins with an occupancy measure that is identical to that of the base policy, and it is strongly penalized for reaching state-action pairs with low or zero probability under $\mu_{\pi\_\text{base}}$, we believe that is it is likely that $\mu_\pi$ will remain absolutely continuous w.r.t. $\mu_{\pi\_\text{base}}$ during training.
>
> **The definition of correlated rewards allows for the correlated reward to deviate arbitrarily in states with zero occupancy by the base policy.** We agree that our definition allows for arbitrarily large differences between the proxy and true reward functions at states that are unreachable by the base policy. However, as we argue above, those states are in practice almost infinitely penalized during training, so we believe that our method will still avoid reward hacking in such cases.

---

> > ### Comment · Reviewer_4snh · 2024-11-27
> >
> > Thank you for your clarification. It addresses my concern.

---

### Official Review · Reviewer_zddp · 2024-11-03

**Soundness:** 3
**Presentation:** 4
**Contribution:** 3
**Rating:** 6
**Confidence:** 4

**Summary:**

The paper addresses the issue of reward hacking in RL, where policies optimized with proxy rewards. To address this, the authors propose a novel, formal definition of reward hacking, which is based on the correlation between proxy and true rewards observed by a base policy. The paper also propose a regularization method based on $\chi^{2}$ divergence between the policies' occupancy measures.

**Strengths:**

The concept presented, while not very complex, demonstrates impressive effectiveness.The authors support their claims with a solid mix of theoretical proofs and comprehensive experiments across various domains. Specific comparison between $\chi^{2}$ vs. KL divergence has been provided. Intuitive examples are also provided. Additionally, intuition about "optimizing the occupancy measure (OM) divergence between policies" might be helpful to the whole community. Overall, this is an excellent paper with good results and insightful contributions.

**Weaknesses:**

However, my primary concern lies in the assumptions underpinning their framework and the choice of metrics. Please see the specific questions below for further details.

**Questions:**

## Questions about underlying assumptions
1. Does the paper assume that both the base policy $\pi_{base}$ and the optimized policy $\pi$ are able to acess the same observation states?
2. **Handling of Unobserved Variables**: In the traffic domain, several potentially impactful variables may be unobserved, as noted in [1][2]. For example, factors like the driver’s expertise level [1] or weather conditions [1] are often missing in driving datasets. How does your framework address scenarios where such unobserved variables exist?
3. During the training phase, does the agent have access to the ground truth reward value?
4. What are major assumptions behind your framework?

## Questions about metrics
1. **Unknown True Reward**: In many practical applications, such as driving, it is challenging to properly define a true reward function [1][2]. In RLHF, the system often relies on preference pairs rather than a true reward. When the true reward is unavailable, what alternative metrics would you recommend for evaluation?
2. **Alternative Metrics for RLHF**: For RLHF, additional metrics might be helpful, such as the winning rate (preferred vs. rejected) or a safety score. How could these be considered in your framework?

## Other concerns
1. **Similarity to Prior Work**: This paper is similar to [3], particularly in the theoretical sections and some experiments. It would be beneficial for the authors to clarify the distinctions between their proposed approach and that of [3].
2. **$\chi^{2}$ vs. KL Divergence**: Does $\chi^{2}$ divergence consistently outperform KL divergence in all scenarios? Are there cases where KL divergence yields better results? If so, what trade-offs are involved? If not, could the authors outline situations where KL divergence might be more suitable?

----------
[1] Ruan, Kangrui, and Xuan Di. "Learning human driving behaviors with sequential causal imitation learning." Proceedings of the AAAI Conference on Artificial Intelligence. Vol. 36. No. 4. 2022.

[2] De Haan, Pim, Dinesh Jayaraman, and Sergey Levine. "Causal confusion in imitation learning." Advances in neural information processing systems 32 (2019).

[3] Laidlaw, Cassidy, Shivam Singhal, and Anca Dragan. "Preventing Reward Hacking with Occupancy Measure Regularization." ICML Workshop on New Frontiers in Learning, Control, and Dynamical Systems.

**Details Of Ethics Concerns:**

Given that this paper focuses on AI alignment, fairness, safety, and privacy (the chosen primary area), with experiments involving RLHF, an ethics review is recommended. Additionally, it would be helpful if the authors could clarify how their proposed method enhances the AI system's safety and fairness.

---

> ### Author Response · Authors · 2024-11-20
> **Rebuttal**
>
> We thank the reviewer for their review and appreciate that they found that our method "demonstrates impressive effectiveness." Below we have responded to individual questions.
>
>  * **Does the paper assume that both the base policy and the optimized policy are able to acess the same observation states?** Yes, we assume that both policies receive the same inputs.
>  * **Handling of unobserved variables in the traffic domain:** We agree with the reviewer that unobserved variables in driving policies trained via imitation learning can cause problems. However, the traffic domain we consider is primarily focused on modeling *traffic*—the high-level interactions of large groups of vehicles—rather than individual drivers. In this case, models of individual driving do not need to be as high-fidelity. Furthermore, the traffic simulator we use is based on SUMO [1], which has been validated for large-scale studies of traffic, such as the VABENE project [2] that was used to inform traffic control during the FIFA World Cup in 2006.
>  * **During the training phase, does the agent have access to the ground truth reward value?** The agents do not have access to the true reward $R$ during training, only the proxy reward $\widetilde{R}$. Our setting is challenging because we evaluate based on the unobserved true reward but only allow policy optimization using a specified proxy reward.
>  * **What are major assumptions behind your framework?** The main assumption that enables our approach to preventing reward hacking is Definition 4.1 in the paper: that the proxy reward is correlated with the true reward over state-action pairs sampled from the base policy. We show that this assumptions holds across four realistic environments and a representative gridworld (see Figure 2).
>  * **Unknown true reward:** We agree with the reviewer that it is difficult to tune hyperparameters, such as the regularization strength $\lambda$, when the true reward is unknown. One option for evaluation in this setting is to learn or specify two different proxy rewards, one for training and one for evaluation. This approach was used for evaluating RLHF-fine-tuned models by Gao et al. [3]. In fact, since the true reward function for RLHF is unknown, in our experiments we leverage a more robust learned reward function as the true reward $R$ and a less robust learned reward function as the proxy reward $\tilde{R}$.
>  * **Alternative metrics for RLHF:** We have additionally computed win rates for both KL and $\chi^2$ regularization in RLHF:
>
>    | Divergence | Median win rate |
> 	 |------------|-----------------|
> 	 | $\\chi^2$  | **52.83**       |
> 	 | KL         | 51.50           |
>
>    To compute the win rates, we compared model outputs from the RLHF-trained models to the SFT baseline using GPT 4o-mini and reported the median rate across five seeds. We see that $\\chi^2$ regularization results in a higher win rate, reflecting our other results. We have included these results in Appendix E.1 of the paper as well.
>  * **Similarity to prior work:** In comparison to the work of Laidlaw et al. [4], we present a more theoretically complete framework and additionally experiment with $\\chi^2$ regularization. In particular, [4] does not have a result like our Theorem 5.1 showing that regularized optimization of the proxy reward can lower bound the true reward. Furthermore, [4] only explores the use of KL divergence for regularization, while we additionally experiment with $\\chi^2$ divergence, which our theoretical results suggest is more principled. We find in practice that regularization with $\\chi^2$ divergence generally outperforms KL regularization.
>  * **$\\chi^2$ vs. KL divergence**: Empirically, we found that across all our experiments, $\\chi^2$ regularization performed similarly to or better than KL divergence. Even when KL divergence performed similarly to $\\chi^2$ divergence, we often found that $\\chi^2$ divergence was more stable across regularization coefficients. Combined with our theoretical results that provide guarantees regarding $\\chi^2$ divergence, this suggests that $\\chi^2$ divergence should be used instead of KL divergence when regularizing to prevent reward hacking.
>
> [1] Krajzewicz, D. (2010). Traffic simulation with SUMO–simulation of urban mobility. Fundamentals of traffic simulation, 269-293.
>
> [2] Krajzewicz et al. (2012). Recent development and applications of SUMO-Simulation of Urban MObility. International journal on advances in systems and measurements.
>
> [3] Gao et al. Scaling Laws for Reward Model Overoptimization. ICML 2023.
>
> [4] Laidlaw et al. Preventing Reward Hacking with Occupancy Measure Regularization. ICML Workshop on New Frontiers in Learning, Control, and Dynamical Systems.

---

> ### Comment · Reviewer_zddp · 2024-11-27
> **Official Comment by Reviewer zddp**
>
> While the initial version of the paper is not perfect, I believe the updated version is better. Especially with the newly added theoretical results, comparing action distribution regularization vs. occupancy measure regularization; and the learned reward can be r-correlated. However, I am uncertain whether adding **ten new pages** is standard practice for such revisions.
>
> Still, regarding the newly added proofs, does it mean that assumption "the proxy reward is **correlated** with the true reward over state-action pairs sampled from the base policy" is required for Theorem 5.1, Corollary A.1, and Lemma A.2? If this assumption is indeed required, it would be beneficial to include experimental evidence or a practical demonstration to substantiate these theoretical claims.
>
> Another concern: Could you elaborate on how these results differ fundamentally from those in [3]. The current proof seems to be easily extended from [3] with Cauchy-Schwartz inequality and commonly known entropy bounds.
>
> Additionally, many common divergences, such as KL-divergence, Symmetric KL divergence, chi-square divergence, Hellinger distance, and total variation distance, are special cases of f-divergence. Why can't we directly use f-divergence here?

---

> > ### Author Response · Authors · 2024-11-29
> >
> > Thank you for responding to our rebuttal. Below we have addressed the points raised in your response:
> >
> >  * **Assumption that the proxy reward is correlated with the true reward:** it is correct that this assumption, which we formalize as Definition 4.1 in the paper, is required for Theorem 5.1, Corollary A.1, and Lemma A.2. We have substantial experimental evidence to support that this assumption is satisfied in realistic environments. First, we show that in four realistic environments and a representative gridworld, the proxy reward is correlated with the true reward, verifying that Definition 4.1 holds (see Figure 2 and the bottom half of page 6). Second, we find that our regularization strategy based on Theorem 5.1 outperforms alternatives in most environments in the experiments in Section 5 (see Table 1), providing further evidence that the assumptions of the theorem are satisfied.
> >
> >  * **Differences from prior work:** there are multiple ways that this paper's results differ significantly from "Preventing Reward Hacking with Occupancy Measure Regularization," the paper referenced by the reviewer. Theoretically, the previous paper only shows that the difference in rewards between two policies is bounded by the total variation divergence in their occupancy measures, assuming a bounded reward function. However, this is a very limited theoretical result; it does not actually show that occupancy measure regularization can be used to *improve* a policy's true reward, only that it can prevent large drops in true reward. Furthermore, we do not assume that the reward function is bounded, making our results more general. Empirically, we test regularized policy optimization on three more environments compared to the previous work and also test $\chi^2$ regularization in addition to KL regularization.
> >
> >  * **f-divergences:** we understand the reviewer's comment to be asking why we don't consider $f$-divergence regularization in general, since the KL and $\chi^2$ divergences are both special cases of $f$-divergences. We consider $\chi^2$ divergence because it is theoretically sound according to Theorem 5.1; the proof of this theorem relies on using $\chi^2$ divergence in particular and does not apply to other $f$-divergences. We also compare KL divergence because it has been used in prior work, particularly on RLHF. There are numerous other $f$-divergences that could be used for regularization but they do not have theoretical justification and have not been used in prior work.
> >
> >    Some of our theoretical results, like Theorem A.5 and Lemma A.6, apply to *any* $f$-divergence. In particular, Theorem A.5 shows that regularizing with any *action distribution* $f$-divergence cannot achieve the same guarantees as Theorem 5.1 achieves with *occupancy measure* regularization. This suggests that we do not need to test other $f$-divergences for action distribution regularization since they are theoretically unsound.
> >
> > We hope that this addresses your concerns. Please let us know if you have any other questions!

---

> ### Comment · Reviewer_zddp · 2024-11-30
> **Official Comment by Reviewer zddp**
>
> > There are numerous other $f$-divergences that could be used for regularization but they do not have theoretical justification and have not been used in prior work.
>
> The claim that these lack theoretical justification or have not been explored in prior work might be somewhat misleading. For reference, here are some important papers on $f$-divergences on RLHF [4][5]. For theoretical justification, see Theorem 1 of [4].
>
> [4] Go, Dongyoung, et al. "Aligning language models with preferences through f-divergence minimization." arXiv preprint arXiv:2302.08215 (2023).
>
> [5] Wang, Chaoqi, et al. "Beyond reverse kl: Generalizing direct preference optimization with diverse divergence constraints." arXiv preprint arXiv:2309.16240 (2023).

---

### Official Review · Reviewer_qUra · 2024-11-04

**Soundness:** 4
**Presentation:** 4
**Contribution:** 2
**Rating:** 6
**Confidence:** 3

**Summary:**

The authors propose a new definition for reward hacking, based on the idea that we should only care about hacking of proxy reward functions which are correlated (in terms of state-action occupancy) with the true reward function under some "reasonable" base policy.

They then show theoretically and empirically that regularizing reward using an occupancy measure term as opposed to the standard KL divergence leads to less reward hacking across several simple environments.

**Strengths:**

The paper is extremely well-written.

The paper builds on past literature in a coherent and helpful way.

The paper shows experiments on interesting environments.

**Weaknesses:**

While the new definition can be defended as an improvement over past definitions, it still leaves some things to be desired. Specifically:

* it requires a "natural base policy", which might not exist in some real-world relevant settings
* even if a natural base policy does exist, it's not obvious that the state-occupancy distribution induced by it is the one we actually care about: the optimal policy wrt the true reward might be quite different from what we think of as the "natural" base policy.
* the definition only considers hacking to have occurred when looking at the *optimal* policy with respect to the proxy; in practice, we never find the optimal policy, yet hacking occurs nonetheless with intermediate sub-optimal (or locally optimal) policies
* the paper doesn't say how to know whether, and what to do if, the (eg, learned) proxy reward function is not r-correlated with the true one.

**Questions:**

## Typos
there is an issue with the text in Definition 4.2: I think your $\tilde{\mathcal{R}}$ ate something after "policy" and another thing before "lower".

## Suggestions
* line 24: "superior" feels a bit strong
* line 164: the SEIR and glucose environments diagrams are too small to read/understand. just a comic of the environment would be better. the gridworld is on the edge of too small but ok
* line 217: I would avoid using the word "misaligned", since that has all kinds of connotations. Consider just "different", "not identical to", etc. This is a nit: if you feel strongly about "misaligned", I think it's probably ok.
* line 312: the SEIR model has been around for a long time (a quick search found stuff from 2013 but I think it's been around for significantly longer); please try to find an original reference OR change the sentence to talk specifically about that one from 2020 which you're using.
* 369: RHS = right hand side? some readers might not know this; please define on first use
* 402: "widely used to prevent reward hacking in RLHF" well, I don't think this is quite true. KL divergence, to my understanding, is primarily used to avoid instabilities in the training procedure (important in when using PPO in general, not just RLHF!). Now, it seems likely that there is a side-effect of this regularization which helps prevent hacking, but that's not the main reason (in particular: if you remove KL divergence, you don't get hacking, you just get a failure to train). Please update this sentence! (or please explain if I'm missing something).

## Questions
* copying from above, my biggest concern is this one: even if a natural base policy does exist, it's not obvious that the state-occupancy distribution induced by it is the one we actually care about: the optimal policy wrt the true reward might be quite different from what we think of as the "natural" base policy. The issue I see here is that if the "actually really good" (much better than "reasonable base") policies have a way different state-occupancy distribution than the base policy, then the "check" that the proxy reward is "similar enough" to the true reward over the "reasonable" distribution just doesn't matter: what matters is what happens near optimality. Am I missing something here?
* why do you only consider the *optimal* policy wrt the proxy in Definition 4.2? This seems too restrictive (and you apply the regularization at all points during training, so unless I'm missing something you don't rely on this optimality?)
* how do you expect your technique to scale?

## Closing note
I gave the paper a 5 because I think it's important that some things be changed before it's ready for publication. But I will be happy to increase the score to a 6 if my minor points are met, and possibly more if my greater fears about the utility of the technique in capturing reward hacking in different but relevant parts of state-action occupancy space are allayed. Indeed, it's possible that just acknowledging and writing about this issue would be enough: so long as it is clearly framed what this paper is and is not doing and to what extent it "covers all the bases" wrt reward hacking, I think it's worth publishing. It's really nicely presented and a nice idea with experiments to back it.

---

> ### Author Response · Authors · 2024-11-20
> **Rebuttal**
>
> We thank the reviewer for their very thorough review and are glad that they believe the paper is “nicely presented and a nice idea.” Below, we have addressed some of their questions and concerns:
>
>   * **Is the state-occupancy distribution induced by a "natural" base policy close enough to optimal?** The reviewer notes that "the optimal policy wrt the true reward might be quite different from what we think of as the 'natural' base policy." We agree that even if the proxy reward is $r$-correlated with the true reward, it could be that the base policy is still quite far from the optimal policy. In this case, it may not be possible to achieve optimal performance under the true reward. However, we argue that this is often not the goal: if all we have is a proxy objective, then it may be unreasonable to expect to achieve *optimal* (or even near-optimal) performance by optimizing the proxy. Instead, as we describe at the beginning of Section 5, we are interested in whether we can "optimize a proxy reward function and have it translate into an improvement in true reward over the base policy." Our Theorem 5.1 shows that this goal of increasing true reward *relative to the base policy* is possible as long as the proxy and true rewards are correlated under the base policy.
>
>     If one cares about finding a near-optimal policy under the true reward, one could combine Theorem 5.1 with an assumption that the base policy achieves reward that is not too far from optimal. We have added such a result as Corollary A.1 in Appendix A.1.1.
>
>   * **Why do you only consider the optimal policy wrt the proxy in Definition 4.2?** We agree that this is a restrictive definition and that in practice the policy is not fully optimized with respect to the proxy reward function. The reason we originally referred to the optimal policy is because it is mathematically precise, while notions of "partially optimizing with respect to a reward function" are harder to formally define. We have updated the definition of "reward hacking" to describe situations when optimizing the proxy (even partially) leads to a decrease in true reward compared to the base policy. Furthermore, a "hackable" proxy reward is one whose optimal policy exhibits reward hacking. Thus, this revised definition now covers cases where the proxy reward is only partially optimized.
>
>   * **The paper doesn't say how to know whether, and what to do if the proxy reward function is not r-correlated with the true one.** We agree that this is a limitation of our paper. While we showed that in several realistic environments that the $r$-correlated proxy assumption is satisfied, this may not be the case in all environments. However, in the case of learned reward functions in particular, we have added Lemma A.8 to Appendix A.4, where we show that under basic assumptions, a learned reward function will be $r$-correlated with the true reward. This gives further theoretical evidence, in addition to our empirical evidence, that many proxies are $r$-correlated with the true reward.
>
>   * **How do you expect your technique to scale?** Since our theoretical results in Theorem 5.1 apply to any method of optimizing the regularized proxy reward on the right side of equation (4), we believe that our technique for preventing reward hacking should scale well to larger models. Furthermore, our experimental results cover both very small networks (e.g., a 2-layer MLP in the pandemic environment) and large language models (Pythia-1.4B for our RLHF experiments). This also suggests that our technique is quite scalable.
>
>   * **KL divergence penalty in PPO:** The reviewer writes that "KL divergence, to my understanding, is primarily used to avoid instabilities in the training procedure [of PPO]." It is certainly true that in PPO there is usually a KL penalty between the current and previous iterations' policies to prevent instability. However, RLHF uses an *additional* KL penalty to avoid reward hacking. While the first KL penalty is between the current and previous PPO policy (which changes each iteration), the second KL penalty is between the current PPO policy and the SFT policy (which does not change). In Stiennon et al. [1], the authors find that removing this second KL penalty leads to overoptimization, producing nonsensical results (see Appendix H.2 of their paper), even though training is still stable (see Figure 5 of their paper).
>
>   * **Other suggestions:** We thank the reviewer for their extensive suggestions about how to clarify our writing. We have incorporated these suggestions in an updated version of the paper uploaded to OpenReview.
>
> [1] Stiennon et al. Learning to summarize from human feedback. NeurIPS 2020.

---

> > ### Comment · Reviewer_qUra · 2024-11-25
> >
> > Thank you for your responses to the weaknesses and my questions.
> >
> > I notice that some of the suggestions I made have not made their way into the current version of the paper. If you think that something isn't a valid concern or there is reason not to do it, could you please let me know? As it stands I assumed they had been included and was surprised to see they weren't.
> >
> > Copying from above:
> > * line 24: "superior" feels a bit strong
> >     * this is still in the paper and appears in several places
> > * line 164: the SEIR and glucose environments diagrams are too small to read/understand. just a comic of the environment would be better. the gridworld is on the edge of too small but ok
> >     * the diagrams haven't changed
> > * line 217: I would avoid using the word "misaligned", since that has all kinds of connotations. Consider just "different", "not identical to", etc. This is a nit: if you feel strongly about "misaligned", I think it's probably ok.
> >     * this is now on line 207?
> > * line 312: the SEIR model has been around for a long time (a quick search found stuff from 2013 but I think it's been around for significantly longer); please try to find an original reference OR change the sentence to talk specifically about that one from 2020 which you're using.
> >     * this is still the 2020 reference
> > * 369: RHS = right hand side? some readers might not know this; please define on first use
> >     * this hasn't been defined
> > * 402: "widely used to prevent reward hacking in RLHF" well, I don't think this is quite true. KL divergence, to my understanding, is primarily used to avoid instabilities in the training procedure (important in when using PPO in general, not just RLHF!). Now, it seems likely that there is a side-effect of this regularization which helps prevent hacking, but that's not the main reason (in particular: if you remove KL divergence, you don't get hacking, you just get a failure to train). Please update this sentence! (or please explain if I'm missing something).
> >     * This one has been addressed, and ideed, it is exactly the KL penalty for RLHF that I was referring to :). I think the phrasing was changed a bit here and I'm now ok with how it's worded. As an aside (no need to change the paper), in the paper you reference, Stiennon et al. say "This KL term serves two purposes. First, it acts as an entropy bonus, encouraging the policy to explore and deterring it from collapsing to a single mode. Second, it ensures the policy doesn’t learn to produce outputs that are too different from those that the reward model has seen during training". Would you argue that the second of these points is them saying "the KL penalty is to mitigate reward hacking"? My best guess is that the KL penalty is doing two things at once: on the one hand, making training more stable, not wanting to diverge too much from the initial distribution (which I see as the main point), and at the same time, possibly preventing some reward hacking (this is motivated by Figure 5 of Stiennon et al., which shows that as KL increases, true reward goes down). Maybe at some point we get into philosophy: what is instability, what is overfitting, and what is hacking?

---

> ### Author Response · Authors · 2024-11-26
>
> Thank you for responding to our rebuttal. We had already addressed a few of the comments you mention, and have now uploaded a new version of the paper with fixes for the remaining ones:
>
>  * **Use of the word "superior":** we used this word as a synonym for "better." However, we understand that it has a stronger connotation, so we have now changed uses of "superior" to "better" or to saying that $\chi^2$ occupancy measure regularization "outperforms" KL action distribution regularization rather than saying it is "superior."
>
>  * **SEIR and glucose diagrams are too small:** we have now changed these diagrams in Figure 2 to be more readable.
>
>  * **Use of the word "misaligned":** we thought that this word was standard in the reward hacking literature [1, 2], but have removed it from the paper.
>
>  * **Reference to the SEIR model:** we removed the reference to Mwalili et al. (2020). We could not find a good original source for the SEIR model—for example, the Mwalili et al. paper does not cite a source for SEIR. It appears that the model [has its origins going back more than a century](https://royalsocietypublishing.org/doi/pdf/10.1098/rspa.1916.0007) and in most SEIR-related papers there is no given citation. Let us know if you believe there is a better way of handling this reference.
>
>  * **RHS acronym:** we had already changed this to define the acronym at its first use: "the right-hand side (RHS) consists of two terms."
>
>  * **Purpose of KL divergence in PPO:** we are glad that the reviewer finds the updated wording better. It's true that Stiennon et al. note that the KL to the SFT policy also serves as an entropy bonus. We do think the point that "it ensures the policy doesn’t learn to produce outputs that are too different from those that the reward model has seen during training" is addressing reward hacking, particularly since without the penalty they get clear reward hacking behavior (see their Appendix H.2). Also, follow-up work to their paper [3] has more clearly shown how KL divergence prevents reward hacking in RLHF. We definitely agree that there is not a clear delineation between "overfitting" to the learned reward function versus "reward hacking" and that stability is closely related as well. Thank you for bringing up this interesting discussion!
>
> [1] Zhuang and Hadfield-Menell. Consequences of Misaligned AI. NeurIPS 2020.
>
> [2] Pan et al. The Effects of Reward Misspecification: Mapping and Mitigating Misaligned Models. ICLR 2022.
>
> [3] Gao et al. Scaling Laws for Reward Model Overoptimization. ICML 2023.

---

> > ### Comment · Reviewer_qUra · 2024-12-03
> >
> > Thank you for your response and for your work during the rebuttal period. I now see no reason this paper should be denied acceptance and have updated my score accordingly.
> >
> > I appreciate the changes. For what it's worth, my main cause for concern was that it sounded like you had addressed them and I was ready to increase my score, but then I took a look at the PDF and found several that it seemed hadn't been addressed in the then-current version. This was quite surprising for me, and felt a bit disingenuous (though I imagine it was more of an oversight / lost while dealing with lots of things on your end), so the alarm bells started ringing in my head and from then on I explicitly wanted them to be acknowledged.
> >
> > Regarding your changes:
> >
> > > superior
> >
> > thank you
> >
> > > diagrams
> >
> > thank you, imo they look much better, I hope you agree!
> >
> > > misaligned
> >
> > Interesting! I'm surprised by the use in the first paper, where it really does seem that they are talking about misalignment between two reward functions. In that case, I guess go ahead and put it back in if you prefer. I (and it seems the second paper you reference) usually think about "misalignment" as something that relates to an _agent_, not to a reward function. The (proxy) reward function can _induce_ misalignment in the agent via training, but I wouldn't call the reward function "misaligned" in itself. For example, in the second paper, they say "Table 1 also indicates whether each proxy leads to misalignment..." which is exactly the kind of way I would talk about it.
> >
> > > SEIR
> >
> > Wow, that's a great reference you found! I'm also not sure what the best option would be here. If other papers don't cite SEIR, maybe it's fine if you don't either? I think that's more "academically honest" than citing a "random" recent paper—I wonder if you would agree.
> >
> > > RHS
> >
> > Sorry for missing this, and thank you for defining it. I don't mean to be overly pedantic about this, but I think it can make a real difference for ESL readers, especially those new to ML and who might not know common English (or ML) acronyms.
> >
> > > KL
> >
> > Thank you too for this interesting discussion, I'll be thinking about it more now in the weeks to come :).
> >
> > Best of luck with your future research.

---

### Author Response · Authors · 2024-11-25
**Summary of rebuttals**

Thank you again to the reviewers for their thorough reviews and helpful feedback. In our rebuttals and in our updated paper we have done our best to address the concerns raised by reviewers, including:

 * We have added over **ten new pages of theoretical results** to Appendix A with references from the main text. These include proving that action distribution (AD) regularization cannot provide the same guarantees as occupancy measure (OM) regularization (responding to Reviewer icTG), showing that learned reward functions are $r$-correlated (Reviewer qUra), investigating whether the lower bound in Theorem 5.1 can be increased above zero (Reviewer icTG), demonstrating a relationship between AD and OM regularization in RLHF where each token is a separate action (Reviewer icTG), and guarantees about near-optimality of policies trained with regularized optimization (Reviewer qUra).

 * We **clarified our paper's relationship** to several other works, including in offline RL, entropy-based regularization for RL, inverse RL, preference-based RL, other work on regularization in RLHF, and the EPIC distance.

 * New empirical results show that $\chi^2$ regularization also **improves RLHF win rates** and that our discriminator-based approximation of occupancy measure ratios is close to theoretically optimal.

As the author-reviewer discussion period is coming to an end, please let us know if there are any other questions that we can help address.

---

### Meta-Review · Area_Chair_AWqd · 2024-12-19

**Metareview:**

The reviewers agreed that the paper investigates an important problem of reward hacking, proposes a novel definition, and then develops a new regularization that can effectively prevent reward hacking. However, the reviewers also raised several concerns and questions in their initial reviews. We want to thank the authors for their responses and active engagement during the discussion phase. The reviewers appreciated the responses, which helped in answering their key questions. The reviewers have an overall positive assessment of the paper, and there is a consensus for acceptance. The reviewers have provided detailed feedback, and we strongly encourage the authors to incorporate this feedback when preparing the final version of the paper.

**Additional Comments On Reviewer Discussion:**

The reviewers raised several concerns and questions in their initial reviews. After the discussion, there was a clear consensus among the reviewers that the paper should be accepted.

---

### Decision · Program_Chairs · 2025-01-22

Accept (Spotlight)